# National climate action can ameliorate, perpetuate, or exacerbate international air pollution inequalities

M. Omar Nawaz [1,2,3] ✉ & Daven K. Henze [3]

Climate action ameliorates public health by reducing hazardous air pollutants alongside greenhouse gases, yet misguided mitigation efforts could induce imbalances in air pollution exchange across international borders. Despite its potential to endanger equality, the effects from climate action on transboundary air pollution are relatively unstudied. Here we show that stricter mitigation increases the fraction of co-benefits that originate externally in Africa by +8% in shared socioeconomic pathways (SSP) towards sustainability (SSP1) and by +53% for fragmentation (SSP3). The fraction of externally originating co-benefits is greater in developing countries (0.76 in SSP1-26) than developed (0.65), indicating that developing countries are more dependent on external action. Although co-benefits are maximized in the most ambitious scenario, SSP1-19 (1.32 million deaths avoided), their transboundary exchange between countries varies. These results suggest a need for climate policies that consider how inequalities in transboundary air pollution evolve across distinct socioeconomic trends and mitigation strategies in addition to total co-benefit estimates.

Climate action has the potential to simultaneously alleviate the global health impacts from climate hazards and reduce the substantial health burden associated with air pollution[1-4]. However, climate policy design that does not account for the geographic distribution of air pollution impacts and transboundary air pollution exchange risks perpetuating or exacerbating global inequalities in air pollution exposure[5,6]. Integrating this consideration in climate policy design is necessary to ensure that current burdens of poor air quality do not continue into the future[7]. With this in mind, the Intergovernmental Panel on Climate Change (IPCC) developed the Shared Socioeconomic Pathways[8] (SSPs) that represent different socioeconomic trends and challenges to mitigation and adaptation that could affect climate change, air pollution, and the inequality of their impacts.

Previous studies of the role of climate action on air pollution have explored how policy implementation could influence air quality and contribute to health impacts globally[2,9], regionally[10,11], and in individual countries[12,13] or cities[14]. These studies have primarily focused on air pollution from particles less than 2.5 micrometers in diameter (PM$_{2.5}$) due to its large health burden[4]. Climate action induces emission changes that affect the transboundary exchange of PM$_{2.5}$, which could heighten or relax international tensions; however, due to the complexity of modeling these interactions, few studies have examined how exchanges in PM$_{2.5}$−and their associated health impacts−could vary across different climate futures. Specifically, considering an individual country, the balance between the benefits realized globally from emission reductions in that country compared to the benefits realized in a country from global emission reductions may differ dramatically across different socioeconomic trends and mitigation strategies. In turn, this could lead to shifting imbalances and inequalities in air pollution exchange. Studies that have explored this question generally demonstrate that policies that deepen global inequalities in emission patterns will subsequently worsen global air quality, especially in

[1]School of Earth and Environmental Sciences, Cardiff University, Cardiff, UK. [2]Department of Environmental and Occupational HealthGeorge Washington University, Washington, DC, USA. [3]Department of Mechanical Engineering, University of Colorado Boulder, Boulder, CO, USA. ✉ e-mail: nawazm3@cardiff.ac.uk

nations classified as developing[15,16]. For example, one study[17] found that improving socioeconomic development was especially important for reducing the burden from poor air quality in low- and middle-income countries as classified based on income levels by the World Bank. In addition to atmospheric transport, trade imbalances in consumption and production also contribute to air pollution inequalities[18,19]. Considering how inequalities in air pollution exchange could evolve across different SSPs for individual countries and regions remains insufficiently explored.

In the coming decades, climate action—or the lack thereof—will shape global efforts to mitigate and adapt to the consequences of climate change and poor air quality[20]. Aggressive and equitable policies could catalyze socioeconomic trends that would ensure that fewer lives are lost from the hazards of climate change while also ameliorating the health burden and inequalities associated with air pollution. In this study, we consider three questions: (1) how might socioeconomic trends (SSPs) and climate mitigations (RCPs) affect the health burdens associated with $PM_{2.5}$ air pollution, (2) how could transboundary exchanges in these health burdens evolve across different climate futures, and (3) how could climate action address or worsen imbalances and inequalities in $PM_{2.5}$-related health impacts. We focus our assessment on the impact of changes in emissions—and not demographics—as they are the most directly controlled by air quality and climate mitigation policy.

To examine these questions, we employ the adjoint of the GEOS-Chem[21,22] chemical transport model (CTM) to conduct simulations that calculate the sensitivity of fine particulate matter ($PM_{2.5}$) exposure (i.e., population-weighted concentrations) in nearly every country of the world to its chemical precursor emissions. We integrate these sensitivities with air pollutant precursor emission projections across a diverse set of socioeconomic and climate mitigation scenarios[23] that are built by coupling SSP and RCP scenarios. We apply health impact assessment methods from the Global Burden of Disease[4] (GBD) 2019 Study and project baseline disease rates and population data into the future from the GBD Foresights[24] project to estimate the health impacts of these scenarios compared to a worst-case scenario (SSP3-Baseline). We identify not only the health impacts from specific scenarios but also—by leveraging the adjoint model source-receptor relationships— how each country contributes to impacts throughout the world in 2040. We compare these two quantities to characterize how transboundary air pollution relationships evolve across different SSPs and to identify how air pollution exposure inequalities could be exacerbated, perpetuated, or ameliorated through different socioeconomic trends. These adjoint modeling results enable us to explore the impacts of climate action across a larger geographic extent (only 25% of these sensitivities have been presented in prior analysis[25]) and allow us to uncover information about climate action in the Global South—i.e., developing nations in Africa, Asia, and Latin America with shared histories of colonialism—a region that has historically been understudied. Crucially, we develop source-receptor matrices for nearly every individual country, facilitating the investigation of policy-related questions throughout the globe without selection or aggregation biases.

## Results

### Climate co-benefits and transboundary inequalities across different climate futures in 2040

Socioeconomic trends (SSPs) and climate mitigation strategies (RCPs) that improve upon the worst-case scenario, SSP3-Baseline, reduce $PM_{2.5}$ precursor emissions in 2040. For example, sustainable development with high climate mitigation (SSP1-19) would lower global nitrogen oxide ($NO_x$) emissions by 73% in 2040. These emission reductions improve air quality and reduce $PM_{2.5}$-related deaths; however, the magnitude of these climate co-benefits varies by scenario (Table 1). In the most optimistic scenario (SSP1-19) in which there is

sustainable socioeconomic development and a strong reduction in climate forcing that presents low challenges to mitigation and adaptation, emission reductions could avoid 1.32 million deaths (with a lower and upper bound of 0.95, 1.73). This is over four times the 0.32 million (0.24, 0.42) deaths avoided in SSP3-60, in which there is fragmented socioeconomic development, weak reductions to climate forcing, and high challenges to mitigation and adaptation. Conventional development (SSP5) and middle-of-the-road development (SSP2) towards RCP-45 result in climate co-benefits between these two extremes of 1.03 (0.75, 1.35) and 0.79 (0.58, 1.04) million deaths avoided, respectively, whereas trends towards inequality (SSP4) for RCP-45 closely match the results of the fragmentation scenario with 0.51 (0.37, 0.67) million deaths avoided.

There are two compounding factors that drive variability across these different climate futures: the degree of mitigation and the type of socioeconomic development. Represented by lower RCPs, stronger mitigation engenders greater co-benefits; however, the strength of this effect is mediated by the socioeconomic development pathway in which the action is implemented. Specifically, the effect of stronger mitigation is relatively weak in sustainable development—yielding 1.1 times as many co-benefits in SSP1-26 compared to SSP1-45—and stronger in more fragmented climate futures—there are 2.0 times as many co-benefits in SSP3-26 compared to SSP3-45. Similarly, socioeconomic development that is more equitable and fosters global cooperation generally leads to greater co-benefits compared to less equitable socioeconomic development, but this is dependent on the degree of climate mitigation. For example, for RCP-26, SSP1 results in 1.4 times as many co-benefits as SSP3 while in RCP-45, SSP1 results in 2.4 times as many co-benefits as SSP3. These results suggest that climate policy would ideally be designed to consider socioeconomic development alongside the degree of mitigation given these dependencies. Ultimately, in climate futures for which there are low challenges to both mitigation and adaptation the improvement to air quality will have substantial health benefits; however, our estimates suggest that challenges to adaptation are especially connected to the deaths avoided from climate action.

Past studies on the impact of climate action on air quality generally estimate the total magnitude of health benefits (as we do in the previous paragraphs); however, this neglects a crucial aspect of climate action with implications for equality: transboundary air pollution. To characterize disparities induced by changes to transboundary air pollution we define two metrics that are associated with air pollution inequality: transboundary fractions and exchanges (Fig. 1). When global climate action is adopted, a fraction of the co-benefits accrued by a country are attributable to external action (i.e., climate mitigation outside of its borders). We define a transboundary fraction (Fig. 1a) that quantifies the fraction of all co-benefits in a country that originate from action outside that country (refer to the "Methods" section for more details). The fraction can also be calculated for regions of aggregated countries. Higher transboundary fractions indicate that a country or region requires more regional or global cooperation to realize the benefits of climate action whereas a lower transboundary fraction positions a country or region as largely being in control of their own climate co-benefits. Transboundary fractions are calculated from the health impacts associated with $PM_{2.5}$-exposure.

We discuss the second metric of exchanges (Fig. 1b) in detail in the next section of the results. For some analyses we group countries into one of six larger regions: South America (SA), Oceania (OC), North America (NA), Europe (EU), Asia (AS), and Africa (AF). The specific countries in each of these regions are provided in Supplementary Table 1 and Supplementary Fig. 1.

For the six regions we consider in this study, scenarios in which climate action is more aggressive (i.e., lower RCPs and SSPs with fewer challenges) achieve greater climate co-benefits (Fig. 2a). In fact, comparing SSP1-19 to SSP3-60, co-benefits are 279%, 410%, 412%, 449%,

**Table 1 | Details on the 24 future climate emission scenarios considered in this study across five shared socioeconomic pathways (SSPs) and six representative concentration pathways (RCPs)**

| SSP | RCP | Global Anthropogenic $NO_x$ emissions (Tg) | Global deaths avoided (millions) [uncertainty] | Narrative | Challenges |
|---|---|---|---|---|---|
| 1 | 19 | 24.0 | 1.32 [0.95, 1.73] | Sustainability | Low challenges to adaptation and mitigation |
| | 26 | 31.8 | 1.26 [0.90, 1.64] | | |
| | 34 | 43.7 | 1.16 [0.83, 1.52] | | |
| | 45 | 52.3 | 1.11 [0.80, 1.45] | | |
| | Ba | 55.4 | 1.08 [0.78, 1.42] | | |
| 2 | 19 | 33.7 | 1.22 [0.88, 1.59] | Middle of the Road | Intermediate challenges to adaptation and mitigation |
| | 26 | 45.8 | 1.13 [0.81, 1.47] | | |
| | 34 | 62.8 | 0.98 [0.71, 1.28] | | |
| | 45 | 85.1 | 0.79 [0.58, 1.04] | | |
| | 60 | 90.5 | 0.75 [0.55, 0.99] | | |
| | Ba | 102.4 | 0.67 [0.49, 0.87] | | |
| 3 | 26 | 61.9 | 0.90 [0.66, 1.17] | Fragmentation | High challenges to adaptation and mitigation |
| | 34 | 80.5 | 0.69 [0.51, 0.90] | | |
| | 45 | 102.1 | 0.46 [0.34, 0.60] | | |
| | 60 | 113.0 | 0.32 [0.24, 0.42] | | |
| 4 | 26 | 57.8 | 0.91 [0.66, 1.18] | Inequality | High challenges to adaptation and low challenges to mitigation |
| | 34 | 79.0 | 0.69 [0.51, 0.91] | | |
| | 45 | 98.1 | 0.51 [0.37, 0.67] | | |
| | Ba | 107.1 | 0.42 [0.30, 0.55] | | |
| 5 | 26 | 33.2 | 1.26 [0.91, 1.64] | Conventional development | Low challenges to adaptation and high challenges to mitigation |
| | 34 | 47.0 | 1.14 [0.82, 1.48] | | |
| | 45 | 58.3 | 1.03 [0.75, 1.35] | | |
| | 60 | 62.0 | 1.00 [0.72, 1.30] | | |
| | Ba | 83.9 | 0.82 [0.60, 1.07] | | |

Global nitrogen oxide ($NO_x$) emissions are presented in teragrams and the global deaths avoided from reductions in $PM_{2.5}$, are presented in millions with lower and upper bound health estimates included in brackets. These refer to anthropogenic emissions and benefits from reductions in anthropogenic emissions, respectively, and so natural $NO_x$ emissions are not reflected in these numbers. The narratives and challenges associated with the SSPs are also included. All scenarios used in our analysis are presented here, with the exception of SSP3-Baseline, which is used as the baseline against which we calculate the climate co-benefits.

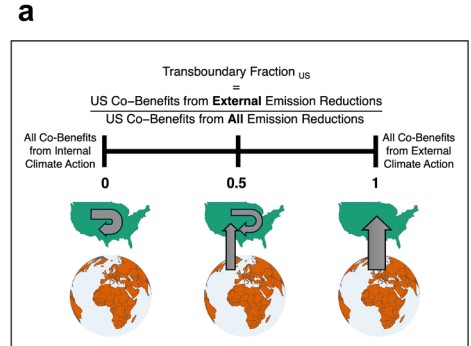

**Fig. 1 | Schematics of the metrics used to quantify inequalities associated with transboundary air pollution across different climate futures. a** Transboundary fractions (TF) characterize the extent of co-benefits in a country that come from external action; co-benefits are the premature deaths avoided owing to reduced $PM_{2.5}$ concentrations. **b** Exchanges (EXC) compare co-benefits from emission reductions between a pair of countries or regions; the contribution to total benefits

exchanged (TEC) indicates how much a country contributes to the total gross co-benefits exchanged between two countries. This Figure was created using Python and the Matplotlib, Cartopy, GeoPandas, and Contextily libraries. Country borders and coastlines are from Natural Earth (public domain), and the basemap is from CartoDB Positron (CC BY 4.0).

548%, and 731% higher in Asia, Oceania, North America, Europe, Africa, and South America, respectively. Regardless of scenario, most of the global co-benefits occur in Asian countries. In SSP1-19, 79% of all co-benefits occur in Asia, placing it far ahead of other regions like Europe (9%), Africa (5%), North America (5%) and South America (3%). In SSP3-60, the Asian share of co-benefits grows (85%) while other regions such

as Europe (7%), Africa (3%), North America (4%), and South America (1%) benefit proportionally less. This high share of co-benefits in Asia is not surprising and is attributable to two factors: (1) the population of Asia is large at present and by 2040 this young population will age and become more susceptible to $PM_{2.5}$-related health impacts; and (2) China and India are developing rapidly and these two countries

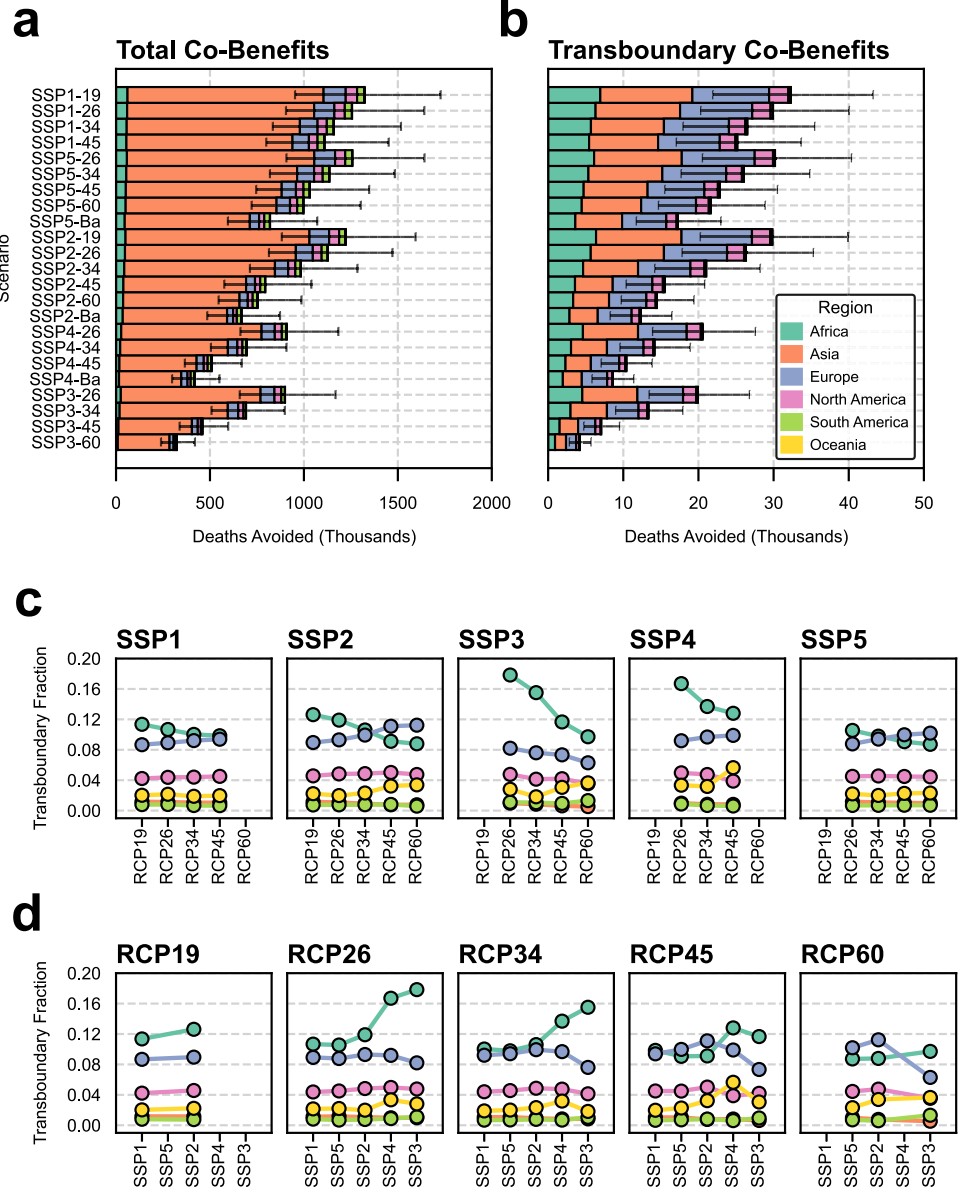

**Fig. 2 | Total and transboundary co-benefits associated with different climate scenarios. a** The total deaths avoided for each of the shared socioeconomic pathway (SSP) and representative concentration pathway (RCP) scenarios relative to SSP3-Baseline in 2040 ordered from the most to least equitable scenarios. Colors indicate the receptor region in which the co-benefits occurred (i.e., where the deaths were avoided). Socioeconomic trends and mitigation strategies are included on the *y*-axis. Error bars refer to the lower and upper bound uncertainty from the health impact assessment for the total co-benefits. **b** The co-benefits specifically attributable to external action for each of the receptor regions (i.e., the transboundary co-benefits). The fraction of co-benefits that are transboundary broken down by **c** socioeconomic development type and **d** mitigation strategy. Data in **a** and **b** are presented as central estimates and lower and upper bounds based on confidence intervals in the health data.

account for a majority of the $PM_{2.5}$-related deaths in the region; thus climate action in these countries could alleviate a great degree of the air pollution health burden. Specifically, co-benefits in these two countries alone make up between 52% (SSP1-19) and 63% (SSP3-60) of all global co-benefits.

Interestingly, although Asia makes up a dominant share of the total co-benefits from climate action, the same is not true for transboundary co-benefits (i.e., co-benefits originating from emissions outside of the region that they benefit) (Fig. 2b). In SSP1-19, Asia has the largest share of transboundary co-benefits (38%); however, this is only slightly ahead of other regions such as Europe (32%) and Africa (21%). In some scenarios, such as SSP1-45, Asia actually makes up a smaller percentage of transboundary co-benefits (32%) than Europe (36%) and only slightly more than Africa (22%) despite its much larger population.

This is an unexpected result. As mentioned previously, rapid development and a large aging population position Asia as the primary beneficiary of climate action; however, this is not the case for transboundary co-benefits. Crucially, the major polluters and population centers of Asia in China and India are located in the eastern part of the continent distant from other regions; thus, a majority (62% in SSP4-45) of the benefits of climate action in the continent are geographically isolated from other regions.

Some regions are more affected by these transboundary co-benefits than others and the influence of external action is sensitive to socioeconomic trends (Fig. 2c) and mitigation strategies (Fig. 2d). Africa is the region most affected by regionally external climate action: on average 12% (9%, 19%) of its co-benefits originate from action in another region. This suggests that African countries are especially

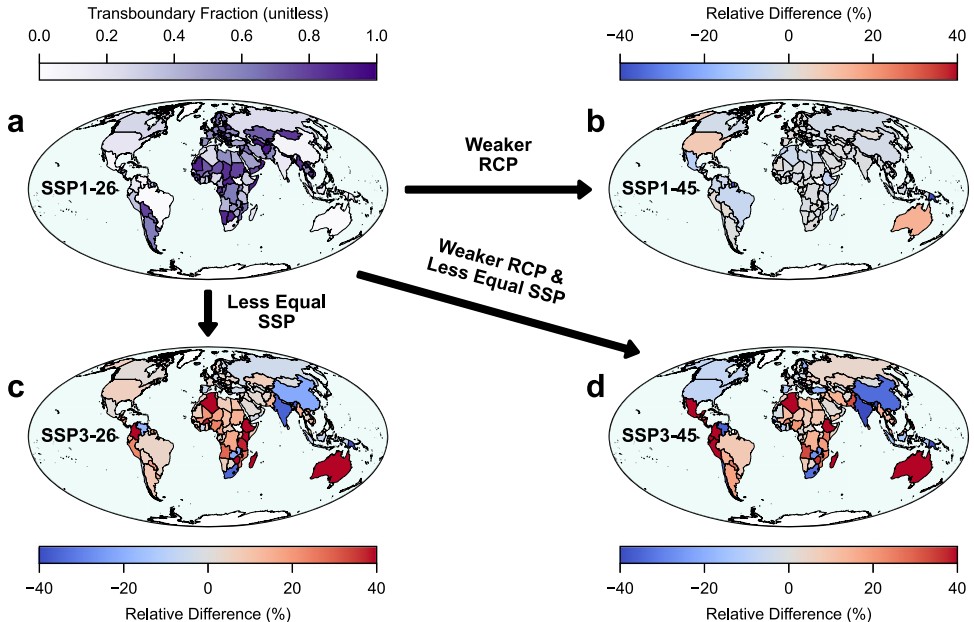

**Fig. 3 | Country-level estimates of transboundary co-benefits across four combinations of shared socioeconomic pathways (SSP) and representative concentration pathways (RCP). a** Transboundary fractions for individual countries in a sustainable and strong climate mitigation scenario (SSP1-26), **b** the relative change in transboundary fractions associated with a weaker RCP, i.e., from SSP1-26 to SSP1-45; red values indicate where transboundary fractions have increased whereas blue values indicate decreases, **c** the relative change in transboundary fractions from a less equal SSP, i.e., from SSP1-26 to SSP3-26, **d** the relative change in transboundary fractions from both a weaker RCP and less equal SSP, i.e., from SSP1-26 to SSP3-45. This Figure was created using Python and the Matplotlib, Cartopy, GeoPandas, and Contextily libraries. Country borders and coastlines are from Natural Earth (public domain), and the basemap is from CartoDB Positron (CC BY 4.0).

reliant on external action to maximize their climate co-benefits. This reliance is mediated by the degree of mitigation action. From RCP-45 to RCP-26, stronger mitigation induces proportionally more transboundary co-benefits across the different socioeconomic development pathways ranging from 8% more in SSP1 to 53% more in SSP3. This relationship between climate mitigation and transboundary co-benefits in Africa is likely attributable to development. The most aggressive climate mitigation scenarios necessitate stronger action in countries that have historically polluted more (e.g., countries in Europe); consequently, as these countries adopt stronger climate action, Africa benefits proportionally more. This is why, across most SSPs, the region with the second highest transboundary fraction—Europe—exhibits an inverse association between climate mitigation and transboundary co-benefits (e.g., a 5% lower transboundary fraction in SSP1-26 compared to SSP1-45). For the other regions, the transboundary fractions are low and on average 5%, 3%, 1%, and 1% for North America, Oceania, Asia, and South America, respectively. In these regions in which major population hubs are more geographically isolated from other regions, domestic climate action dominates co-benefits.

Surprisingly, less equal socioeconomic development increases the transboundary fraction for Africa. One reason why this could occur is that in more fragmented climate futures, there is less cooperative development. Consequently, in these less equal scenarios, African countries become even more sensitive to global action (67% higher in SSP3-26 than SSP1-26) as there is less local development and industrialization. This relationship is true across all of the RCPs; however, it is stronger in those with higher climate mitigation (i.e., RCP-26) than those with weaker mitigation. Specifically, comparing SSP3-45 to SSP1-45, the transboundary fraction is only 18% higher. Ultimately, the transboundary fraction in Africa is mediated by two competing phenomena: (1) more aggressive climate mitigation necessitates proportionally more involvement from developed nations and thus higher transboundary co-benefits and (2) less equal socioeconomic development further increases the disparity in industrialization and

development in many African countries which forces them to be more dependent on external action to maximize co-benefits.

Transboundary fractions exhibit greater variability for countries than regions: e.g., there is a standard deviation of 0.29 for countries compared to the 0.04 for regions in SSP1-26. This variability in the transboundary fraction is influenced by the degree of development (Fig. 3). For SSP1-26, the transboundary fraction for the 20 countries with the lowest Human Development Index (HDI)[26] is on average 0.76 compared to 0.65 for the 20 highest HDI countries. Additionally, 90% of these low HDI countries are African nations with the two exceptions being Afghanistan (0.91) and Yemen (0.78). In contrast, the 20 countries with the highest HDI are primarily from Europe, North America, Oceania, and Asia; none of these countries are from Africa or South America. This suggests that developed nations receive proportionally fewer of their co-benefits from external action than developing nations and consequently that global cooperation benefits developing nations more than developed ones. The total co-benefit numbers are also included (Supplementary Fig. 2) to distinguish areas of high and low impact when interpreting the transboundary fractions.

There are several underlying factors driving the transboundary fractions: e.g., proximity to major polluters, geographic size, prevailing winds, and population density. Island nations, and those isolated by geographic features (e.g., Chile) tend to have more domestic pollution than nations proximate to high polluters (e.g., much of Europe). In the Northern Hemisphere, prevailing westerlies lead to more pollution transport to the East of major polluters in comparison to the West. Thus, it is important to consider that transboundary fractions can indicate equitable or inequitable policy outcomes when considered alongside the HDI of a country (Supplementary Fig. 3). Generally, countries with high transboundary fractions require more compensatory actions as they have less control of their air pollution burden, whereas countries with low transboundary fractions can adopt action more independently. However, given that geography and population density influence transboundary fractions, it is crucial to consider a

dimension of development alongside this exchange. The greatest targets for compensatory action are those countries with high transboundary fractions and a low level of development. These countries will likely bear heavier health burdens in the future as industrialization and technological progress occur.

The global transboundary fraction (i.e., the sum of all external co-benefits over all co-benefits) is much lower than for most individual countries and it ranges from 0.23 (SSP3-60) to 0.28 (SSP1-19). However, this fraction is largely brought down from the inclusion of India and China who receive most of their co-benefits from internal action. For example, in SSP3-60 removing these two countries leads to a higher global transboundary fraction of 0.54. This explains why when calculating the average transboundary fraction across all countries there is a substantially higher value of 0.68 (SSP3-60) than the total global transboundary fraction. We note that these transboundary contributions are substantially greater than past analyses[27,28] of historical transboundary air pollution burdens, which found 14% and 12% of all $PM_{2.5}$-related deaths were attributable to transboundary pollution. This difference is likely attributable to three main factors: (1) these past studies focused on aggregated groups of countries whose boundaries are further from peak emissions locations whereas we perform this calculation for individual countries, (2) in this study we consider climate action whereas these previous studies consider all anthropogenic emissions; climate action is not adopted to the same degree globally and many of the highest emission reductions are implemented in a small set of developed nations, and (3) many of the targets of climate action are precursors of secondary $PM_{2.5}$ (specifically $NO_X$ and $SO_2$) and not primary $PM_{2.5}$ and thus this action targets pollution that is more likely to be transported greater distances due to its longer atmospheric lifetime.

These inequalities associated with transboundary air pollution are dependent on the socioeconomic development and climate mitigation future in which they take place. Considering HDI again, transboundary fractions decrease slightly for the 20 lowest HDI countries (−1%) while increasing substantially for the 20 highest (+32%) when transitioning from SSP1-26 to SSP1-45. In the inverse scenario, in which mitigation is held constant (RCP-26), but the socioeconomic development becomes less equal (from SSP1 to SSP3), transboundary fractions increase in the 20 highest HDI countries (+3%) and even more so in the 20 lowest HDI countries (+22%). This is consistent with our previous results at the regional scale and is likely attributable to the fact that stronger mitigation necessitates greater participation from developed nations and climate action taken in less equal socioeconomic development increases transboundary co-benefits in the developing world because they are more dependent on action outside of their borders due to weaker internal development and industrialization.

Compounding weaker mitigation (RCP-45) with less equal socioeconomic development (SSP3) does not exhibit additive effects of the two individually. For example, while Mexico is only slightly affected by changes to mitigation (−10%) and socioeconomic development (+1%) alone, their compounded effect greatly increases the transboundary fraction (+53%) indicating more contributions from external action than in SSP1-19. Thus, this again suggests the need to consider climate mitigation within the socioeconomic development framework for which it is implemented in.

The main results presented are for 2040; however, we additionally calculate the differences in transboundary fractions and co-benefits between the base year of study (2040) and 2030 (Supplementary Figs. 4 and 5) across SSP1-26 and SSP3-26. We find that in 2040 transboundary fractions are moderately higher in China, India, the US, and Brazil and lower in parts of Africa compared to SSP1 in 2030; however, these interannual changes are smaller than those attributable to socioeconomic changes. We additionally project changes in transboundary fractions from

2030 to 2050 across the set of scenarios (Supplementary Fig. 6) to identify regional-scale trends. Generally, transboundary fractions remain stable through time; however, in more equal SSPs (i.e., SSP1), transboundary fractions for Africa generally decrease from 2030 to 2050 (e.g., by 15% from 2030 to 2050 in SSP1-45) compared to less equal SSPs (i.e., SSP3) where transboundary fractions increase (e.g., increase by 30% from 2040 to 2050 in SSP3-45). We note that our estimates of co-benefits for 2030 lie outside of the lower-bound health estimate for 2040 indicating that the difference in these estimates is not captured by uncertainty in the health calculation alone.

Ultimately, these findings imply two outcomes: (1) the effects of climate action on transboundary air pollution are unique to socioeconomic development and mitigation strategies and these effects are not additive and (2) less developed countries receive greater transboundary co-benefits in less equal socioeconomic development as they are more dependent on external action due to weaker internal development and industrialization.

## Climate co-benefit exchanges from transboundary air pollution

The exchange of climate co-benefits between pairs of countries (or regions) evolves in response to different socioeconomic development pathways and climate mitigation strategies. We characterize the intraregional and interregional exchange of climate co-benefits by calculating exchange metrics (Fig. 1b) that represent the air pollution co-benefits realized in one country that are attributable to climate action in another country. In our analyses, we consider exchanges in three ways: (1) in their absolute sense an exchange represents how emission reduction in one country induces co-benefits in another, (2) as net exchanges by differencing the two exchanges in a pair, and (3) the proportion of co-benefits contributed by one member of a pair to the total gross co-benefits exchanged between the pair to determine if a specific country dominates the exchange of air pollution co-benefits. In the latter approach, we note that exchanges in which there is a greater disparity—especially when the low HDI member of the pair is contributing more than the higher one—are key targets for compensatory action.

For most country pairs, one country contributes more co-benefits to the other than vice versa, leading to inequalities in exchange even in optimistic scenarios such as SSP1-26 (Fig. 4). Considering only exchanges with more than 30 co-benefits, for Europe, exchanges vary from being relatively balanced such as Germany-Sweden (1.06 times higher co-benefit contribution from Germany than Sweden), Russia-Romania (1.14), UK-Netherlands (1.14), and Ukraine-Poland (1.15) to very imbalanced such as Spain-Portugal (4.64), Czechia-Germany (4.58), and France-Spain (4.13). For Africa the most balanced exchange in SSP1-26 is between South Africa and Mozambique (1.27) and the most extreme is between Uganda-DRC (28.5) in which the effects of higher population, prevailing easterly trade winds, and a greater degree of development compound to contribute to substantial transboundary air pollution exchange inequality. For Asia, Pakistan-India (1.01), Thailand-Philippines (1.03), Thailand-Vietnam (1.11), and Pakistan-China (1.11) are the most balanced in SSP1-26 while China-Vietnam (21.0), Thailand-Myanmar (16.4), and India-Myanmar (12.8) are the most imbalanced. Between the two largest players in Asia, China and India, China contributes 1.8 times as many co-benefits to India as vice versa.

These exchange inequalities can be exacerbated or ameliorated through different socioeconomic development pathways. For Africa, fragmented socioeconomic development transforms the relatively equivalent exchange between South Africa and Mozambique in SSP1-26 to one in which South Africa contributes much more (3.39 times). For Asia, the previously equivalent Pakistan-India exchange in SSP1-26 has greater contributions from India (1.33) in SSP3-26. The inequal exchange between China and India is perpetuated in middle-of-the-

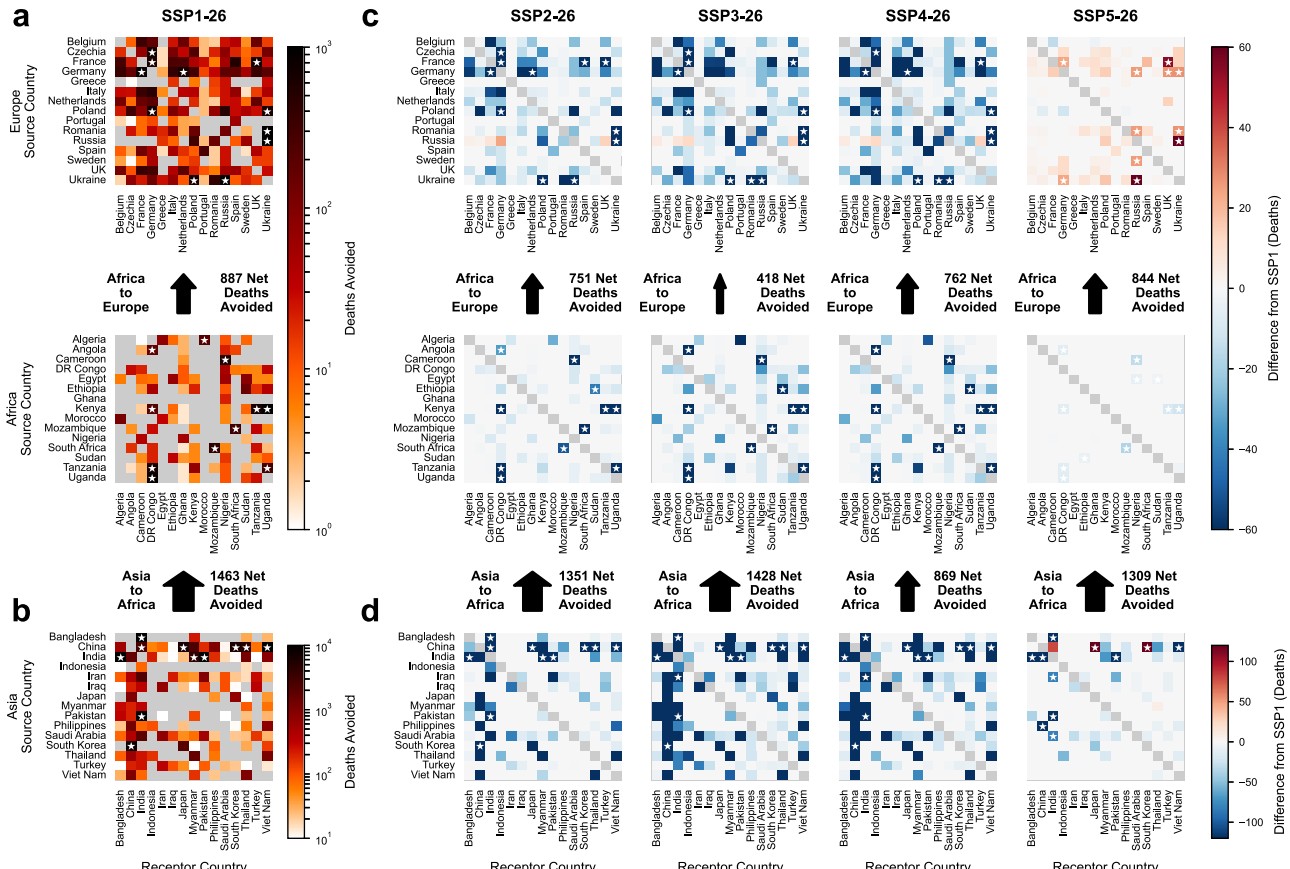

**Fig. 4 | Exchanges (EXC) of climate co-benefits within and between different regions across the five shared socioeconomic pathways (SSP). a** Exchanges for Europe and Africa, for SSP1-26 in 2040 (left). Contributions are indicated through a logarithmic colormap ranging from 1 to 1000 deaths avoided. Darker colors indicate greater co-benefits in the receptor country attributable to emission reductions in the source country; self-contributions (i.e., the diagonal) are excluded. **b** EXC within Asia and between Asia and Africa; here contributions are indicated in a logarithmic colormap that ranges from 10 to 10,000 deaths avoided. **c** Heatmaps of transboundary exchanges of climate action within and between Europe and Africa in 2040 for the scenarios SSP2-26, SSP3-26, SSP4-26, and SSP5-26 relative to SSP1-26. The colormaps are linear and range from −60 to +60 fewer or more deaths avoided compared to SSP1. **d** Heatmaps of transboundary exchanges of climate action within Asia and between Asia and Africa in 2040 for the scenarios SSP2-26, SSP3-26, SSP4-26, and SSP5-26 relative to SSP1-26. The colormaps are linear and range from −120 to +120 fewer or more deaths avoided compared to SSP1. For all subplots, interregional exchanges (i.e., Africa to Europe and Asia to Africa) are provided in an absolute sense—not relative to SSP1. White stars are placed to indicate the top 5% highest (absolute) values for each heatmap.

road (1.79), fragmented (1.76), and inequality (1.82) development pathways; however, it notably increases through conventional development (2.07). Lastly, in Europe, fragmented development exacerbates the exchange inequality between Spain and Portugal (5.78) but partially reduces the inequality between France and Spain (3.67). Many of the European co-benefit exchanges increase in conventional development; this is likely owing to higher climate mitigation (RCP-26) paired with conventional development (SSP5) which necessitates greater climate action in developed countries (including many in Europe) compared to other socioeconomic pathways in which there is more even and sustainable development. When considering weaker climate mitigation as we include in Supplementary Figs. 7 and 8, we see that many of these increased exchanges are resolved.

Exchange inequalities between regions are also sensitive to the socioeconomic pathway in which the climate action takes place. In sustainable development, African countries contribute 887 more co-benefits to Europe than vice-versa; however, in fragmentation this dynamic weakens, and African countries instead contribute fewer (418), albeit still more, co-benefits to Europe than vice versa. The exchange between Asia and Africa is consistently imbalanced towards Asia: in sustainable development Asia contributes 1463 more co-benefits to Africa than vice versa; however, this net exchange decreases to 869 co-benefits in socioeconomic trends towards inequality.

Ultimately, socioeconomic development dramatically affects inter-regional and intraregional exchange inequalities; however, this effect is not uniform. The same socioeconomic trend can simultaneously work to ameliorate some exchange inequalities while perpetuating or exacerbating others.

Comparing fragmented socioeconomic development to sustainable socioeconomic development results in some of the largest changes to exchange inequalities between regional and country pairs. For example, in SSP3-45, the exchange between Europe and Africa is relatively balanced: the former contributes 48% of the total co-benefits exchanged and the latter, 52% (Fig. 5a). Transitioning to more sustainable socioeconomic development leads to Africa—the region with the lower GDP today (as classified by the World Bank)—contributing a clear majority of co-benefits to the exchange (72%). This shift towards greater African contributions to Europe in a more sustainable world occurs because a more sustainable climate future necessitates growth in historically underdeveloped regions (i.e., countries with HDI < 0.7) and enables them to contribute more to climate action and subsequently greater contributions to co-benefits. Contrasting this with the transition from a middle-of-the-road to sustainable climate future (Fig. 5c), the exchange between Africa and Europe is essentially unaffected (from 29% to 28%, African favored). This suggests that fragmentation contributes to a balanced Africa and Europe exchange and

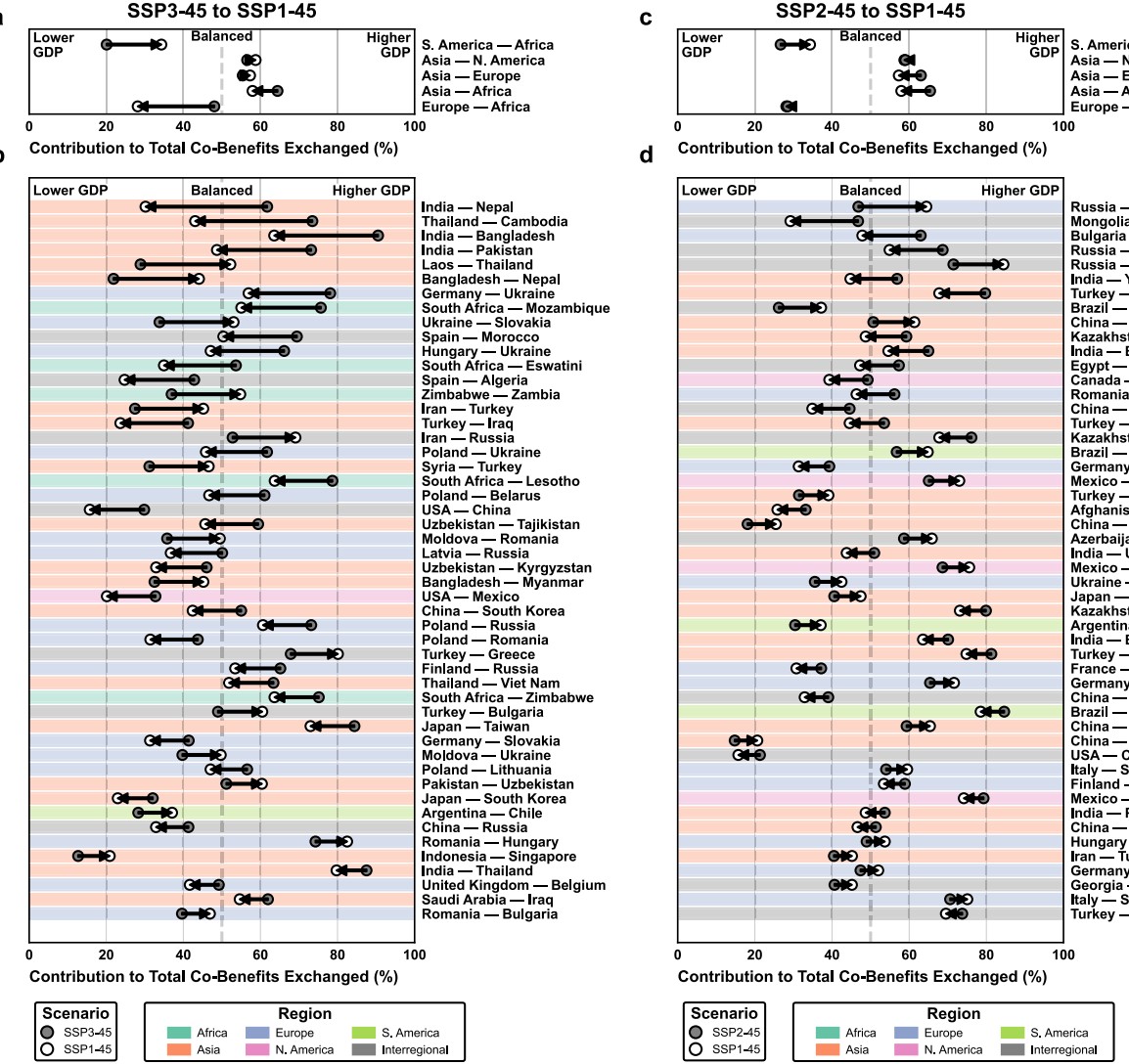

**Fig. 5 | The change in the percentage of co-benefits exchanged (TEC × 100%) between regional and country pairs that is contributed by the higher GDP (first name).** For the regional exchanges from the shared socioeconomic pathway (SSP) scenario SSP3-45 to SSP1-45 in 2040 (**a**), the arrow points from the less equal scenario (SSP3-45) to the more equal scenario (SSP1-45) and the gray dot represents SSP3-45 and the white dot represents SSP1-45. For the country exchanges (**b**), the arrow and dots are the same but the background shading indicates in which region the exchange occurs as labeled in the legend below the subplots; gray indicates exchange between different regions. **c**, **d** are the same as (**a**) and (**b**), respectively, but for middle-of-the-road development (SSP2 vs. SSP1) instead of fragmentation development.

through trends towards sustainable or middle-of-the-road development, Africa contributes more to Europe than vice versa due to a greater capacity to adopt more aggressive climate action. Generally, at the regional scale, changes to exchanges were mostly unaffected by this socioeconomic development excluding the exchange between South America and Africa which becomes less African dominated in sustainable development compared to both SSP3-45 and SSP2-45.

There is even greater sensitivity to socioeconomic development at the country-scale (Fig. 5b). When transitioning from fragmented (SSP3-45) to sustainable (SSP1-45) development there is generally more balance between country exchange pairs especially in the pairs that undergo the largest change to exchanges: in the top ten pairs, the country with the higher current GDP (as classified by the World Bank) at present contributed 61% of co-benefits in SSP3-45 compared to 50% in SSP1-45. Nine of these ten pairs had more balanced exchange patterns (i.e., closer to 50%) in sustainable development futures compared to fragmentation futures. Crucially, sustainable development consistently pushed towards greater contributions from the lower GDP member of the exchange pair, ranging from a 32% lower TEC in the

India-Nepal pair to 19% lower in Spain-Morocco. This supports previous findings that more sustainable futures will enable greater contributions to co-benefits from countries with lower GDPs due to greater development and industrialization. In contrast, in the transition from SSP2-45 to SSP1-45 (Fig. 5d), this trend towards more balance does not occur; average contributions from the higher GDP change only from 57% in SSP2-45 to 54% in SSP1-45 in the top 10 nations.

Another distinction in transitioning to SSP1-45 from SSP3-45 versus SSP2-45 is the regions in which the largest changes to exchange patterns occur. When comparing fragmentation to sustainable development, 7 of the 15 largest exchange shifts occur in Asia with the remaining 8 occurring in Africa (3), Europe (3), or interregional exchanges between Spain and North Africa (2). In contrast, when comparing the middle-of-the road scenario to sustainable development, only one of the largest exchanges includes an African country, not just in the top 15 but the entire top 50 (i.e., the Egypt-Ukraine exchange). Beyond this, the top 15 is largely made up of Asian (5), interregional (6), and European (3) exchanges. This suggests that fragmented socioeconomic development especially affects African

exchanges compared to other socioeconomic development and exchange inequalities can be ameliorated by transitioning towards more sustainable development. The degree of climate forcing (Supplementary Fig. 9) also has implications for regional exchanges. In both sustainable development and fragmented development, the pairs that are most affected by the difference between an RCP-45 and RCP-26 future are largely those pairs that include developed nations in Europe and Asia. This supports our earlier finding that more aggressive climate mitigation requires more contributions from developed nations.

## Discussion

We integrate adjoint sensitivities with SSP and RCP emission projections to extend beyond past work that estimated total co-benefits from climate action and consider how transboundary co-benefits and exchange relationships between countries and regions could change across different climate scenarios in 2040. We introduce capabilities for characterizing how country-specific source-receptor relationships worldwide change in response to different challenges to mitigation and adaptation. This enables us to investigate how climate action could affect inequalities in the global distribution and exchange of air pollution that progresses beyond prior analyses that focused on quantifying the magnitude of benefits from emission reductions or examined aggregated[28] or limited[25] source receptor relationships.

We estimate that socioeconomic trends and mitigation strategies have the capacity to avoid between 0.32 (0.24, 0.42) million (SSP3-60) and 1.32 (0.95, 1.73) million (SSP1-19) deaths compared to a worst-case scenario (SSP3-Baseline) in 2040. These co-benefits are largely concentrated in Asia, and China and India especially (between 52% and 63%) suggesting that climate action in these countries—regardless of the socioeconomic pathway—is needed to maximize co-benefits and minimize intraregional and interregional inequalities. Specifically, this is more beneficial for ameliorating intraregional inequalities in Asia, as Asian countries only receive between 32% and 38% of all transboundary co-benefits and they are closely followed by Europe (32% to 36%), and Africa (21% to 22%) despite their substantially smaller populations. This proportionally weaker transboundary influence in Asia is a product of population density: major populations in Africa and Europe are in closer proximity to extraregional polluters, whereas in Asia most of the high population is concentrated in the Eastern part of the continent, away from other regions. Thus, climate action with implications for interregional transboundary air pollution will proportionally be more important for Africa and Europe. Interestingly, we note that the fraction of co-benefits that are external at the country-scale (on average 0.68 in SSP3-60) is generally larger than past analyses[27,28] of historical transboundary burdens that found 14% and 12% of air pollution-related health impacts were transboundary. This can be explained through the fact that climate action induces air pollutant emission precursor reductions distinct from total emission patterns historically and today. Thus, the air pollution co-benefits from climate action in many countries appear to be disproportionately external; additionally, climate action tends to target the longer-lived precursors of secondary $PM_{2.5}$ (as opposed to primary $PM_{2.5}$) and thus has greater transboundary implications.

For both regions and countries, the fraction of co-benefits that are transboundary is closely tied to the scale of climate mitigation and the socioeconomic development pathway in which the action takes place. African countries exhibit higher transboundary fractions in high mitigation scenarios and less equal socioeconomic development. The former relationship occurs because high mitigation scenarios require proportionally more action from more economically developed areas. The latter relationship is due to the fact that more fragmented socioeconomic development induces greater development disparities and subsequently more global dependence on external action throughout many countries in Africa and other developing areas of the world. This implies a need for nuanced climate policy design that incorporates

climate mitigation action in tandem with projected socioeconomic development to ensure more equitable climate co-benefits.

Considering HDI, many of the lowest development countries have higher transboundary co-benefits in SSP1-26 whereas the highest HDI countries tend to have lower transboundary co-benefits. This is unsurprising, as development and industrialization increase so too does air pollution and thus in countries with lower HDI there is less capacity for climate co-benefits and more co-benefits originate externally. However, this positions these low HDI countries in a disadvantaged position in which they are dependent on global or regional action to maximize their co-benefits. Notably, these fractions increase through fragmented socioeconomic development and thus inequal development can further stress this relationship.

Exchanges between pairs of countries or regions are imbalanced across different socioeconomic development pathways. For example, in Africa in a sustainable future (SSP1-26), South Africa and Mozambique contribute relatively equal co-benefits to one another (South Africa contributes 1.27 times as many); however, in a fragmented future (SSP3-26) the number nearly triples (3.39). Across the same socioeconomic transition, the imbalance between China and India is relatively unchanged: China contributes 1.8 times as many co-benefits to India as vice-versa in SSP1-26 compared to 1.76 in SSP3-26; interestingly, conventional development exacerbates this imbalance to 2.07. The transition to more sustainable development does not always reduce inequality, e.g., for France and Spain the exchange in SSP3-26 is more balanced (3.67) than in SSP1-26 (4.13). Ultimately, there is not a uniform relationship between socioeconomic development and exchange inequalities: the same transition could ameliorate some exchange inequalities while perpetuating or exacerbating others. However, generally a transition from a more fragmented climate future to a more sustainable ones lead to greater balance: comparing SSP1-45 to SSP3-45, nine of the ten largest shifts in exchanges between countries enabled more balanced exchanges (i.e., where pairs of countries contribute even co-benefits to one another) and many of these shifts were towards greater contributions from the lower GDP country. This occurs because sustainable development fosters greater global cooperation; developing nations develop and industrialize quicker and thus they have more potential to control emissions.

The results presented in our work should be considered alongside sources of uncertainty. First, our adjoint-derived methodology calculates linear relationships between emissions and $PM_{2.5}$ and while some components of $PM_{2.5}$ (e.g., primary carbonaceous aerosol) are formed linearly, others (e.g., secondary inorganic aerosol) are formed through complex non-linear chemistry[29]; previous work[30] has suggested that this non-linearity could lead to underestimates of $PM_{2.5}$-related health impacts up to 57%. Additionally, our estimates of co-benefit contributions rely on the simplified assumption that the pollution formation relationships that we calculate for our base year (i.e., 2010) will be maintained in the future. We anticipate that changes to climate[31,32] and chemical[33,34] environment in the future will modify transboundary air pollution and we suggest that feedbacks between climate change and transboundary air pollution are explored in future work. Our analysis neglects to include anthropogenic fugitive dust and secondary organic aerosol formation that contribute to $PM_{2.5}$, especially in heavily populated urban areas[35]. Previous studies[36,37] indicate that for urban locations, between 5 and 10% of $PM_{2.5}$ originates from non-local SOA. We suggest that future work is done to quantify the transboundary fractions of SOA globally. Another limitation is that this analysis is conducted for emissions scenarios in a single year—2040—so it represents a snapshot in time of the differences between these socioeconomic trends and mitigation strategies. Given that the SSP-RCP emissions are designed to peak at different points across different countries and regions the results presented in this work should be considered only as a snapshot of the inequalities in the year 2040 as the transboundary fractions and exchanges will differ at different

points in the future. In this study we do not explore the evolution of inequalities in detail; however, a benefit of this adjoint methodology is its ability to rapidly assess different years−as well as different scenarios −offline without additional simulations. To demonstrate this, we include figures for an additional year (i.e., 2030) in Supplementary Figs. 10–13. Additionally, we compare the transboundary fractions and co-benefits in 2030 to those in 2040 (Supplementary Figs. 4 and 5) at the national-scale and estimate the regional trends in transboundary fractions from 2030 to 2050 (Supplementary Fig. 6); these additional results suggest that less equal SSPs likely will exacerbate transboundary fractions through time more than more equal ones. Lastly, although we estimate uncertainty associated with the health calculation, there is additional uncertainty in the projected changes to population, age distribution, and baseline disease rates that are not captured in our analysis along with uncertainty in the air quality modeling that we discuss in detail in the methods.

With these uncertainties in mind, our results have relevance for climate policy design. First, the specific inequality metrics we define, transboundary fractions and exchanges, have implications for policy. In quantifying the fraction of co-benefits that are transboundary, countries and groups can identify the scenarios that are most beneficial in specifically alleviating transboundary air pollution external to their borders. Through quantifying exchanges between countries or regions, they can identify the most important partnerships to address imbalanced exchanges of air pollution. Second, there is evidence from our results that socioeconomic development and mitigation action are best considered in parallel rather than separately; the socioeconomic environment in which mitigation occurs dramatically influences both the total magnitude of co-benefits and the inequalities associated with transboundary air pollution. More fragmented socioeconomic development increases transboundary air pollution in developing countries (especially in Africa) and makes them more beholden to the decision-making of developed countries. Stronger climate mitigation, although overall beneficial, also increases the fraction of co-benefits that are transboundary as the most aggressive climate action requires greater buy-in from historically polluting developed nations. From a socioeconomic development perspective, transitioning from fragmented to sustainable climate futures leads to more balanced exchanges of co-benefits which enables developing countries to contribute proportionally more to exchanges due to greater industrialization and development. Ultimately, this suggests that sustainable socioeconomic development enables developing countries to participate more in global climate action, thus benefitting both themselves and their wealthier and more developed neighbors via improved domestic and foreign air quality.

Overall, transboundary air pollution presents a massive health burden and represents a source of global environmental inequalities today. Sustainable climate action has the capacity to improve public health and reduce global inequalities; however, nuanced climate policy design that incorporates climate mitigation alongside socioeconomic development is encouraged to represent the inequality impacts associated with transboundary air pollution.

## Methods

To assess the health benefits of socioeconomic trends and mitigation strategies, we integrate emission scenarios, adjoint modeling, and health data. First, we perform adjoint sensitivity calculations for nearly every country in the world to characterize the sensitivity of $PM_{2.5}$ exposure to emissions of its precursors. Then, we combine these sensitivities with a consistent set of projected SSP and RCP emissions[23] for 2040 to estimate how $PM_{2.5}$ exposure in these countries could change as a result of projected changes in emissions. We then estimate the health impacts associated with changes in $PM_{2.5}$ exposure for each country following established methods[4] from the GBD 2019 study. Additionally, we use data from the GBD Foresights

project[24] to project population and baseline disease rates into the future.

### Emission scenarios for socioeconomic trends and mitigation strategies

We use a consistent set of gridded global emissions data that were generated using an integrated assessment framework[23]. These data cover multiple socioeconomic assumptions (i.e., SSP1–SSP5), climate mitigation levels (RCP 1.9 Wm$^{-2}$, 2.6 Wm$^{-2}$, 3.4 Wm$^{-2}$, 4.5 Wm$^{-2}$, 6 Wm$^{-2}$, and Baseline), and chemical species including black carbon (BC), ammonia ($NH_3$), nitrogen oxides ($NO_x$), organic carbon (OC), and sulfur dioxide ($SO_2$). Emissions are provided for 2005 and then every ten years from 2010 to 2100; we extract the 2040 emissions for this study and also use the 2030 emissions for additional supplemental analyses. Emissions are separated by sectoral source in the dataset; we aggregate the emissions across these sectors excluding those associated with biomass burning (i.e., agricultural waste burning, deforestation, savanna burning). We include the following sectors: agriculture, aviation, residential and commercial, power plants, industry, international shipping, solvents, surface transportation, and waste. These emissions are generated at the 0.5° × 0.5° resolution; we generate country masks at this resolution from SEDAC CIESIN[38] and, for each country, regrid the emissions to the model resolution (2° × 2.5°) using the conservative regridding algorithm from the xESMF python library[39].

### Air quality and adjoint modeling

We simulate the formation of $PM_{2.5}$ using the GEOS-Chem[22] chemical transport model, specifically, version 35 of the adjoint[21]. We conduct global simulations at a horizontal resolution of 2° × 2.5° with 47 vertical layers driven by GEOS-5 assimilated meteorology for 2010 from the National Aeronautics and Space Administration (NASA), Global Modeling and Assimilation Office (GMAO)[40]. Our simulations contain two components: first, we perform the forward model simulation in which model sensitivities are propagated forward in time from emissions to form $PM_{2.5}$, second, we perform the adjoint calculation in which a reverse integration of the forward model is calculated to estimate the sensitivity of scalar cost-functions of country-scale population-weighted $PM_{2.5}$ to precursor emissions. We perform the simulations in 2-month increments run in parallel as discussed in our prior work[25]. Overall, we perform these coupled calculations (i.e., the forward and adjoint simulations) for 168 countries amounting to around 2000 total 2-month simulations. This version of GEOS-Chem does not include fugitive dust and secondary organic aerosol which biases our results low as discussed in our prior work[25] and in the uncertainty analysis section.

To define the cost-functions of our adjoint simulation that are relevant for health outcomes, we integrate satellite-remote sensing derived estimates of $PM_{2.5}$ exposure[41] to downscale and rescale our simulated concentrations[36,42] to reduce resolution-based uncertainty in our health impact assessment. These data are available at 0.1° × 0.1° and correspond to a global population-weighted $PM_{2.5}$ concentration of 32.6 µg m$^{-3}$ with a variance of up to 33.9 µg m$^{-3}$ in parts of Asia and Latin America, where uncertainty was highest. Additionally, we calculate the population-weighted average of these downscaled concentrations for each country to characterize the $PM_{2.5}$ exposure (i.e., how populations are exposed to $PM_{2.5}$). We use the cost-function definition for $PM_{2.5}$ from our prior work[36] and calculate sensitivities to BC, OC, $NH_3$, $NO_x$, and $SO_2$ emissions. After calculating the cost-functions, the adjoint simulation calculates the linear sensitivity of population-weighted $PM_{2.5}$ exposure in a country to these emissions:

$$\lambda_{I,k,m} = \nabla_{E_{I,k}} J_m = \frac{\partial J_m}{\partial E_{I,k}} \tag{1}$$

The adjoint sensitivities ($\lambda_{I,k,m}$) of population-weighted $PM_{2.5}$ in country $m$ to emissions from grid cell ($I$) and precursor species ($k$) are calculated by taking the gradient of the cost-function ($J_m$) for country $m$. We employ adjoint simulations, as opposed to finite difference calculations from standard forward model simulations, to be more computationally efficient. To calculate the sensitivities for $PM_{2.5}$ exposure for each country and scenario for a single year would require 48,900 2-month simulations; this is over twenty times as many simulations as our adjoint approach. As employed here, the adjoint sensitivities are limited in that they represent the linear response to the cost-function from the model parameters (i.e., emissions), second order and non-linear effects are not captured through this approach. While the emission response of primary $PM_{2.5}$ is linear, there are still substantial non-linear effects of secondary inorganic aerosol that are not captured using this adjoint approach that are investigated in more detail in prior work[25].

### Projecting changes in $PM_{2.5}$ exposure from emission scenarios
We combine the gridded emission projections from the SSP and RCP scenarios[23] with the gridded adjoint sensitivities for each country and each chemical precursor species, to estimate how projected changes in emissions in all grid cells contribute to changes in $PM_{2.5}$ exposure for each receptor country:

$$\Delta J_{I,m,s} = \sum_k \lambda_{I,k,m} \Delta E_{I,k,s} \tag{2}$$

$$\Delta E_{I,k,s} = E_{I,k,s} - E_{I,k,\text{SSP3-Baseline}} \tag{3}$$

where $\Delta J_{I,m,s}$ is the contribution to the cost-function—i.e., $PM_{2.5}$ exposure in country $m$—from emissions in a $2° \times 2.5°$ grid cell, $I$ for country $m$ and scenario $s$. Specifically, we consider the emission delta, $\Delta E_{I,k,s}$, that is the difference between emissions from a scenario ($s$) $E_{I,k,s}$ compared to emissions from the SSP3-Baseline scenario in a chosen year (for this study, primarily 2040) $E_{I,k,\text{SSP3-Baseline}}$. We note that we selected 2040 as the main year for our analysis because it was the first year in which SSP3 scenarios began to deviate from the SSP3-Baseline, it was the latest year for which we had GBD Foresights data (see next section), and approximately when global $NO_x$ emissions peak for the SSP3-Baseline scenario. In calculating these contributions to the cost function we can identify how emissions from every grid cell contribute to $PM_{2.5}$ exposure in a receptor country and—using country masking data from SEDAC CIESIN[38]—identify how changes in emissions in specific countries or country blocs could influence air pollution in the future for each receptor country. Further details on this methodology can be found in our prior work[14,25,30,36,42–46].

### Health impact calculation
We calculate the health impacts associated with changes in annual average $PM_{2.5}$ exposure associated from emission scenarios relative to the SSP3-Baseline following established methods[25]. Briefly, we use relative risk tables developed in the GBD 2019 study[4] to relate specific $PM_{2.5}$ exposures to increased risks of premature death from ischemic heart disease, stroke, chronic obstructive pulmonary disorder, acute lower respiratory illness, lung cancer, and type-2 diabetes. We combine these relative risks with national baseline disease rates and population data from the GBD 2019 study for a base year of 2010—the same as the adjoint sensitivities. For future years (i.e., 2040) we project these population and disease rates using results from the GBD Foresight project[24] as we established in our prior study[25]. Premature deaths for country $m$ are estimated as:

$$\text{Mortality}_m = y0_m \left(1 - \frac{1}{RR_m}\right) \text{Pop}_m \tag{4}$$

Where $y0_m$ corresponds to the national baseline disease rate in country $m$ projected to the year of analysis (i.e., 2040), $RR_m$ refers to the relative risk derived from a $PM_{2.5}$ exposure in country $m$ and $\text{Pop}_m$ corresponds to the population in country $m$ projected to the year of analysis. For each grid cell, country, and scenario we perform this calculation twice: once calculating $RR_m$ using the baseline $PM_{2.5}$ exposure from the cost-function and a second time with the contribution ($\Delta J_{I,m,s}$) removed from the cost-function. By taking the difference of these two premature deaths estimates we calculate the number of deaths avoided—relative to the SSP3-Baseline—from emission reductions in every grid cell ($I$), SSP/RCP scenario ($s$), and in each receptor country ($m$). We note that we perform this calculation for each distinct combination of health outcomes and age groups, following the GBD methodology, and then aggregate these estimates. Ultimately, this approach identifies how emissions from specific locations and scenarios affect every country for which we perform an adjoint sensitivity calculation.

### Calculating transboundary fractions and exchanges in air pollution health impacts
To explore how inequalities in transboundary air pollution—and the associated co-benefits from climate emission scenarios—vary across different climate futures, we define a set of metrics. First, we consider the transboundary fraction:

$$\text{TF}_{m,s} = \frac{\sum_{I \notin m} \Delta J_{I,m,s}}{\sum_I \Delta J_{I,m,s}} \tag{5}$$

The transboundary fraction ($\text{TF}_{m,s}$) for country $m$ and scenario $s$ is equivalent to the ratio of the co-benefits contributed to country $m$ from emissions outside of the country compared to the co-benefits contributed from everywhere, including the country. This is calculated by considering emission reductions that occur in grid cells that are not within $m$ ($I \notin m$), over the benefits in country $m$ contributed by emission reductions from all grid cells. A transboundary fraction close to one indicates that country $m$ receives most of its benefits through external emission reductions, whereas a ratio close to zero indicates that domestic emission reductions contribute to more benefits within $m$ than reductions elsewhere. Next, we consider exchanges:

$$\text{EXC}_{m,n,s} = \sum_{I \in m} \Delta J_{I,n,s} \tag{6}$$

The exchange ($\text{EXC}_{m,n,s}$) between countries or regions $m$ and $n$ for scenario $s$ is equivalent to the amount of air pollution-related benefits in $n$ that are attributable to emission reductions in $m$ for scenario $s$. The net exchange refers to the difference between two exchanges in a country pair (i.e., $\text{EXC}_{m,n,s}$ - $\text{EXC}_{n,m,s}$). Lastly, the contribution to total co-benefits exchanged, used in Fig. 5, refers to the percentage of total co-benefits exchanged between a pair attributable to one member of the pair:

$$\text{TEC}_{m,n,s} = \frac{\text{EXC}_{m,n,s}}{\text{EXC}_{m,n,s} + \text{EXC}_{n,m,s}} \tag{7}$$

Thus, the contribution to total co-benefits exchanged, $\text{TEC}_{m,n,s}$, compares the exchange contributed by country $m$ to country $n$ ($\text{EXC}_{m,n,s}$) to the total gross co-benefits exchanged between the two countries ($\text{EXC}_{m,n,s} + \text{EXC}_{n,m,s}$). A TEC value of 0.5 indicates balanced contributions to co-benefits exchanged between countries, while values above 0.5 indicate greater contributions from country $m$ and values below 0.5 indicate greater contributions from country $n$. Schematic depictions of these metrics are included Fig. 1.

## Uncertainty analysis

Uncertainty is introduced primarily by: (1) emission projections, (2) the GEOS-Chem forward simulation, (3) the adjoint sensitivity calculation, and (4) the health impact analysis.

Although it is impossible to quantify a specific uncertainty associated with emissions projected for the future—given that the true values are unknown—the emission projections[23] were tested against historical emissions from another inventory, the Community Emissions Data System (CEDS)[47] and they were found to be well correlated for $SO_2$ from all sectors in 2005 ($R^2 = 0.71$). Additionally, the correlation with CEDS projections in 2050 for the energy sector was examined for both $SO_2$ ($R^2 = 0.44$) and $NO_x$ ($R^2 = 0.70$). The HTAP emissions[48] that drive our forward simulation have been compared to other inventories[49]. Of the 12 regions considered in that analysis, HTAP had the minimum emissions of all bottom-up inventories in five of the regions for $NO_x$ (27% lower than average), none of the regions for $SO_2$, five of the regions for BC (44% lower than average), and three of the regions for OC (48% lower than average). Additionally, HTAP had the maximum emissions in none of the regions for $NO_x$, two of the regions for $SO_2$ (39% higher than average), two of the regions for BC (21% higher than average), and two of the regions for OC (62% higher than average). This comparison suggests that the HTAP emissions have regional biases; however, compared to CEDS, the HTAP emissions of $NO_x$ are approximately 16% lower on a global scale. Given the degree of uncertainty in these emission projections, we exclusively consider the relative differences in projections as opposed to their absolute implications for our analysis.

GEOS-Chem has a strong track record of estimating atmospheric composition that compares well to observations[50,51]. In an HTAP ensemble analysis[52] of the $PM_{2.5}$ health burden associated with intercontinental transport, the simulated $PM_{2.5}$ of the GEOS-Chem adjoint was compared to over 3000 global monitors. GEOS-Chem adjoint simulated concentrations fell within the range of ensemble members with a normalized mean error of 55.1% (ranged from NME = 35.4% to NME = 62.9%) and a correlation of 0.65 (ranged from $R = 0.63$ to $R = 0.77$). The GEOS-Chem adjoint had the most positive bias of the ensemble members (NMB = 20.3%); however, in an absolute sense the bias was in the middle of the spread of ensemble members (ranged from NMB = −60.9% to NMB = 20.3%). Additionally, all of the ensemble members underestimated the health burden of $PM_{2.5}$ compared to the GBD 2015 study (4.2 million premature deaths); thus, this potential high bias in the GEOS-Chem adjoint led to a more accurate estimate of health impacts (3.2 million premature deaths) than the multi-model mean (2.8 million premature deaths). We also note that a majority of the surface-level observations were located in North America, Europe, and China; thus, this analysis may not be representative of biases in other regions of the world.

The version of GEOS-Chem that is used in our analysis does not include anthropogenic fugitive dust and secondary organic aerosol; the prior was estimated[53] to increase global population-weighted $PM_{2.5}$ by around $2.9 \, \mu g \, m^{-3}$ and the latter was estimated[35] to make up between 15–30% of anthropogenic $PM_{2.5}$ in urban environments. Fugitive dust $PM_{2.5}$ is a primary species with a short atmospheric lifetime that mainly affects concentrations near where it is emitted. In contrast, secondary organic aerosol has a longer atmospheric lifetime enabling its transport across greater distances. For example, one study[37] simulated $PM_{2.5}$ transport for one UK city, and found that approximately 5% of the total $PM_{2.5}$ came from non-local secondary organic aerosol. In our previous work[36], we estimated that around 10% of $PM_{2.5}$ in DC, originated from non-local secondary organic aerosol. These past analyses suggest that the exclusion of SOA could lead to an underestimate of between 5–10% of non-local $PM_{2.5}$ that in turn would imply an underestimate of transboundary fractions and exchanges by this magnitude. However, we note that these past studies are for two locations in the Global North and not representative of the regional

variation in this study and that the SOA contributions are calculated for urban areas that have higher local sources than rural areas.

Another source of uncertainty in our analysis is the application of local-linear sensitivities to inherently non-linear $PM_{2.5}$ formation. Although the primary components of $PM_{2.5}$ (i.e., black and organic carbonaceous aerosol) respond linearly to emissions, secondary inorganic aerosol does not respond linearly. Thus, this approach does not capture the second order effects of secondary inorganic aerosol. A previous study[30] explored the uncertainty associated with this local-linear assumption in response to SSP emission projections in Korea, and found that non-linearities contributed an underestimate of up to 57% in the $PM_{2.5}$-related health benefits from mitigation (although this uncertainty was lower for 2040).

Lastly, we quantify the uncertainty introduced from the health impact analysis by using upper and lower bound estimates of the different components of the health calculation (i.e., the relative risk, population estimates, and baseline disease rates) using the lower and upper bound values provided by the GBD study. This uncertainty is the only value included in the uncertainty bounds presented throughout the text given that previous work suggests that this is the largest contributor to uncertainty in our approach[54]. We do not consider uncertainty in the projected changes to health data (i.e., the projected population and disease rates for 2040).

### Reporting summary

Further information on research design is available in the Nature Portfolio Reporting Summary linked to this article.

## Data availability

The emission projection data used in this study are publicly available for free from Fujimori et al. and accessible at: https://www.nature.com/articles/sdata2018210. The baseline disease rates, relative risk data, and population data are available from the Global Burden of Disease study (https://vizhub.healthdata.org/gbd-results/). The GEOS-Chem adjoint sensitivity calculations are available from the corresponding author upon request. The source data used to generate all of the main figures for this article and all source-receptor health estimates across the scenarios are freely available for open access on Zenodo (https://zenodo.org/records/18008107).

## Code availability

The GEOS-Chem adjoint source code used to calculate adjoint sensitivities is publicly available, and instructions for downloading it can be found on the GEOS-Chem Adjoint Wiki: https://wiki.seas.harvard.edu/geos-chem/index.php/GEOS-Chem_Adjoint_Model.

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

## Acknowledgements

The authors acknowledge funding support from both NASA NNX16AQ19G and 80NSSC19K0193 for both M.O.N. and D.K.H. We additionally acknowledge funding support from Cardiff University's Open Access fund for covering the article processing charges for this publication.

## Author contributions

M.O.N. and D.K.H. conceived and designed the study. M.O.N. and D.K.H.performed the simulations and processed the data. M.O.N. analyzed the data and developed the figures. M.O.N. wrote the initial draft of the manuscript. M.O.N. and D.K.H. contributed to the interpretation of the results, revised the manuscript, and approved the final version for submission.

## Competing interests

The authors declare no competing interests.
