## [Transparent Peer Review file · Nature Communications]

National climate action can ameliorate, perpetuate, or exacerbate international air pollution inequalities

Corresponding Author: Dr Muhammad Nawaz

Version 0:

Reviewer comments:

Reviewer #1

(Remarks to the Author)

The manuscript evaluates how socioeconomic trends (SSPs) and climate mitigations (RCPs) might affect the health burdens associated with PM_{2.5} air pollution, and explores PM_{2.5} transboundary exchanges and associated environmental inequities across different climate scenarios. This manuscript is well organized and provides interesting results. However, several issues need to be addressed before the manuscript can be considered for publication.

(1) The study involves the coupling of multiple data and models, including GEOS-Chem, Health impact calculation, and so on. The authors should consider the uncertainty introduced by each link and its impact on the final results to increase the credibility.

(2) The PM_{2.5} transboundary exchanges include atmospheric transport and trade exchange, as should be noted in the text. And there are some studies that have revealed the impact of atmospheric transport on health and environmental inequities in historical years that the authors can compare the findings of this paper with them.

(3) The manuscript defines two metrics to describe unfairness, the imbalance ratio and the exchange ratio, but their descriptions in the main text are too difficult to understand, such as Line 538-555, Line 54-56. The authors' intent is not understood until Figure S1.

(4) Based on the impacts of climate action on PM_{2.5}-related health and inequity analyzed in this paper, the authors can make specific recommendations for climate pathways or mitigation actions in future years.

(5) What is the basis for the article's choice of 2040 as the study year? The results for 2030 are also presented in SI; is there any difference in the findings on inequality between the two years?

(6) Confidence intervals are missing for some of the premature deaths.

(7) What are the data sources for baseline incidence and population in 2040? national scale?

(8) The font size of Figure 1 (a) (b) and Figure 3 (a) is too small. Consider enlarging or using abbreviations.

Reviewer #2

(Remarks to the Author)

Review Comments for NCOMMS-25-17271:

While the manuscript presents a comprehensive assessment of transboundary PM_{2.5}-related health co-benefits under various climate futures, and introduces metrics such as the imbalance ratio and exchange ratio, the overall methodological innovation is incremental rather than groundbreaking. The study builds upon previously published adjoint-based sensitivity analyses, extending the scale and scope, but without introducing novel modeling frameworks or substantially new insights.

Major concerns:

1. The methodology is largely based on previous work. The modeling approach, including the use of the GEOS-Chem adjoint model for estimating sensitivities of PM_{2.5} exposure to precursor emissions, has been described in several earlier studies. The integration with health impact assessment methods and SSP-RCP emissions scenarios also follows a structure that has previously been applied. In the current manuscript, the methodology is scaled to a global level and applied to a broader set of scenarios. However, no substantial changes to the model, workflow, or conceptual framework are introduced. As such, the novelty of the methodology appears to be limited. For a journal that prioritizes conceptual or methodological advancement, this may not be sufficient.

2. While the imbalance and exchange ratios are introduced as new indicators, they are derived directly from existing model outputs and are relatively straightforward in formulation. Their interpretation remains largely descriptive, and their implications are not developed in depth. The study shows that some regions receive more co-benefits than they contribute and that these patterns vary across climate scenarios. However, these observations tend to confirm existing assumptions about global inequalities rather than reveal new or unexpected findings. It is not shown how these indicators would be useful for designing future policy or guiding negotiations on climate action and air quality management.

3. All results are presented for the year 2040. Since SSP-RCP pathways evolve differently over time, it is unclear whether the identified patterns of imbalance and exchange are consistent across other time points. A single-year analysis may be influenced by scenario-specific assumptions in that year and does not allow for understanding temporal trends or cumulative effects. Although additional figures for 2030 are provided in the supplement, the main analysis remains focused on one year, which limits the robustness of the conclusions.

4. The version of GEOS-Chem used in this study does not include secondary organic aerosol formation, which can contribute significantly to PM_{2.5}, especially in urban and vegetated regions. The lack of these components may lead to an underestimation of long-range transport and influence the imbalance and exchange results. Additionally, the adjoint sensitivities are based on linear approximations, which may not fully capture the nonlinear chemical reactions involved in PM_{2.5} formation, particularly in areas with high emissions of ammonia and nitrogen oxides. The limitations of the chemical representation are acknowledged briefly in the methods section, but their implications for the main findings are not discussed in detail.

5. Although the manuscript touches on environmental justice and potential inequalities in the distribution of air pollution benefits, it is not made clear how the presented metrics could be applied in practice. There is a lack of connection between the model outputs and current international climate mechanisms or policy instruments. The imbalance ratio, in particular, is discussed without a clear normative interpretation. It remains uncertain whether a high or low ratio should be viewed as desirable or problematic, and how such findings could inform equitable burden sharing in climate action.

Specific Comments on Figures, Tables, and Technical Details:

1. In lines 99–101, the main text states that SSP1 avoids 0.53 million deaths, but Table 1 reports 581,000 (i.e., 0.581 million) deaths for SSP1-19. While the values are close, the correspondence between the text and specific scenarios in Table 1 should be clarified to avoid confusion.

2. The concept of the imbalance ratio is introduced around line 131, but its calculation is not explained until the Methods section (line 531 onward), and the visual schematic appears only in the supplement (Figure S1a).

3. In Figure S5, the “Asia to Europe” exchange ratio under one of the SSP-RCP scenarios (likely around SSP2-34 or SSP2-45) spikes to a value exceeding 5, which is visually inconsistent with the rest of the dataset and other scenario curves. This point is not acknowledged or explained in the figure caption, nor is there any indication that the y-axis was clipped or rescaled to accommodate outliers.

4. The caption for Figure 4 explains that values greater than 1 mean Region A contributes more to Region B’s benefits than vice versa. However, this phrasing could be misread as “Region A receives more benefit” rather than “Region A’s actions have greater effect”. Rephrase the caption to clarify directionality, e.g., “Values greater than one indicate that climate mitigation in Region A provides more benefits to Region B than the reverse.”

5. Table 1 presents NO_x emissions in teragrams and avoided deaths in thousands, but these units are not explicitly listed in the table headers. I think add units in parentheses to the headers for clarity (e.g., “Global NO_x Emissions (Tg)”, “Global Deaths Avoided (Thousands)”) will improve current version.

6. Oceania is excluded from Figure 1c on the grounds of having disproportionately high imbalance ratios due to small population size. However, no supporting data is shown to validate this exclusion. Please provide numerical values or variance ranges for Oceania in the supplement to justify its removal from the main analysis.

Reviewer #3

(Remarks to the Author)

Reviewer #4

(Remarks to the Author)

This study involves a substantial amount of work. It presents regional inequality in the benefits of emission reduction under different SSP and RCP scenarios from a novel perspective, while also considering transboundary pollution transport. However, several issues still require the authors’ attention. Please make major revisions to the article.

Comments:

1. The abstract is difficult to understand without reading the full manuscript. It requires appropriate revision for clarity.
2. Table 1 does not present all scenarios, such as SSP3-Ba. The authors should clarify the reason for this omission.
3. The reference to Figure 1 in line 116 seems inappropriate. In addition to providing Table S1, a spatial distribution map of the six regions should also be included.
4. Why are Europe and Africa grouped into the same category? Is this classification unrelated to socioeconomic development levels? The rationale should be further explained.
5. What does "IR" refer to in line 223? Its full name should be provided upon first appearance.
6. What is the reason for the inclusion of the four specific regions in Figure 2a? Please clarify.
7. The results presented in the manuscript are relatively limited. It is recommended to include spatial distribution and flow maps of health co-benefits under different scenarios.
8. The manuscript primarily provides descriptive explanations of the observed phenomena, but lacks further analysis of the underlying causes. What specific factors lead to regional inequality in climate action benefits under different scenarios? Are these related to economic levels or population structures? Moreover, are there any suggestions for more equitable mitigation scenarios or compensatory mechanisms?

Version 1:

Reviewer comments:

Reviewer #1

(Remarks to the Author)

I appreciate the authors' detailed response, and most of the issues have been adequately addressed. Regarding the uncertainty in the results, the authors acknowledge in their discussion that uncertainties arise from various aspects such as emission projections, chemical transport model simulations, and adjoint sensitivity calculations. However, the current revision still limits the quantification of uncertainty mainly to the health impact calculations. In the GEOS-Chem analysis, simply stating that "transboundary pollution may be underestimated" is insufficient. Readers may want to know what this implies for the final estimates of mortality.

This work does not present major innovations about the methodological framework. Extensive literature has already discussed the uncertainty of chemical transport models and the propagation of model uncertainties. It is suggested that the authors refer to some methodological studies to provide a more substantive quantification of the uncertainties in their results. And, there is a minor issue that the equation on line 725 appears to be incomplete.

Reviewer #2

(Remarks to the Author)

The figures play a central role in communicating the results; however, I have identified several issues regarding figure formatting, consistency, and clarity that require substantial revision prior to publication. Addressing these issues will substantially improve the quality and impact of the visual presentation in this manuscript. Below, I provide detailed comments for each figure:

General Figure Issues:

1. There is a lack of consistency in font style and size across figures, with some using Arial and others Times New Roman.
2. Legends and labels are sometimes incomplete or formatted in a non-standard way.
3. Axis labels, panel annotations, and units are not always clearly presented.
4. In-figure legends and explanations are missing in some multi-panel figures, making interpretation less accessible.

Specific Figure Comments:

Figure 2: 1. All textual elements, including axis labels, titles, legends, and color bars, are set in Times New Roman, while Figures 1, 4, and 5 use Arial. I recommend standardizing all figures to use Arial and adopting consistent font sizes. For example, use 14 pt for titles and 12 pt for axis labels and legends throughout the manuscript; 2. The color legend for continents in panels (a) and (b) does not follow standard formatting. Please revise using standard legend boxes with clear labels for each region; 3. The x-axis in panel (c) is incomplete, as only partial scenario labels such as 19 and 26 are shown. This may be confusing to readers. Please ensure that all axis labels and tick marks are fully represented.

Figure 3: 1. All elements in this figure are presented in Times New Roman. For consistency, please change all fonts to Arial so that all figures follow the same style; 2. The upper color bar does not include a unit. Please add "Transboundary Fraction (unitless)" or "Transboundary Fraction (%)" as appropriate. For the lower color bars, make sure the unit "percentage (%)" is clearly indicated; 3. Currently, the figure does not display panel labels or scenario flow arrows. I recommend adding clear panel labels, such as (a), (b), (c), and (d), in the upper left of each map. It would also be helpful to include arrows to illustrate the direction of scenario changes; 4. The figure caption should be expanded to provide a distinct explanation for each panel, including a description of the scenarios being compared and the direction of change.

Figure 4: 1. I recommend restructuring the figure legend so that each panel "(a), (b), etc." is described separately. For each heatmap, the legend should clearly state the main variable displayed, the numerical range or color scale, and the main findings or notable patterns. Moreover, heatmaps can be improved through the addition of markers (e.g., bold boxes, stars) to

identify top contributors or outliers.

Figure 5: 1. The font size for country names and axis labels is too small. Please increase font size and consider bolding labels for clarity; 2. Please add explicit in-panel legends explaining the meaning of black/white dots and arrow directions, rather than relying solely on the main caption.

Reviewer #3

(Remarks to the Author)

Reviewer #4

(Remarks to the Author)

Manuscript number: NCOMMS-25-17271A

Title: National climate action can ameliorate, perpetuate, or exacerbate international air pollution inequalities by Nawaz et al.

The authors have made thorough and thoughtful revisions in response to the initial round of comments. The manuscript has been significantly improved through the incorporation of additional analyses, clarification of methodologies, and enhanced discussion of results and implications. The introduction of “transboundary fractions” and “exchanges” as well as the inclusion of uncertainty analyses strengthen the paper’s contributions to understanding how climate action may affect international air pollution inequalities.

Despite these substantial improvements, several issues require further attention to fully improve the manuscript’s quality.

(1)The focus on 2040 is well-justified, but the analysis remains a single-time-slice comparison. The supplemental analysis of 2030 is appreciated, but the manuscript should more explicitly discuss the limitations of not capturing temporal evolution, especially since emission pathways and socioeconomic assumptions evolve complexly. A brief discussion on how key inequalities might trend over time (e.g., towards 2050) would strengthen the manuscript.

(2)The revised metrics (transboundary fractions and exchanges) are much clearer now. I suggest the authors present some detailed discussions about their implications for climate justice or policy design. The authors might also more explicitly link metric values to equitable policy outcomes, e.g., whether certain values indicate need for compensation, cooperation, or independent action.

Line 22. Abstract should distinguish between background and original results. To improve clarity of the abstract, please explicitly introduce key results with phrases such as “our results reveal that...” or “we find that...”.

Line 63-64. The sentence on atmospheric transport and trade is somewhat abrupt. Consider integrating it more smoothly or expanding with a phrase like “In addition to atmospheric transport, trade imbalances also contribute...”

Line 148-160. Please clarify whether this fraction is calculated based on mass of PM_{2.5} or health impacts (deaths).

Line 251-261. The correlation between transboundary fraction and HDI is insightful. Have the authors considered also controlling for geographic size or population density? Smaller or coastal countries may have inherently different transboundary fractions regardless of development status.

Line 303-308. The comparison between 2030 and 2040 is useful. Please include in the main text whether the differences are statistically significant or within uncertainty bounds.

Line 530–553. The uncertainty paragraphs in the Methods and Discussion are somewhat repetitive. Consider rephrasing or cross-referencing to avoid duplication.

Figure 3. The maps are effective, but I suggest consider adding a panel showing the absolute number of transboundary co-benefits (deaths avoided) in addition to the fraction. This would help distinguish between high-fraction but low-impact regions vs. high-impact regions

The authors have done a great job revising the manuscript and addressing most of the initial concerns. The paper presents a valuable and timely analysis of how climate action may affect international air pollution inequalities. With the additional clarifications and discussions suggested above, the manuscript will be suitable for publication in Nature Communications.

Version 2:

Reviewer comments:

Reviewer #1

(Remarks to the Author)

We appreciate the authors' thorough additions and revisions addressing the quantification of uncertainties. By citing multiple relevant studies in the revised manuscript, the authors have provided a quantitative discussion of uncertainties in key aspects such as emission inventories, model simulations, and health impact assessments, significantly enhancing the rigor and transparency of the analysis. We believe the robustness of the results has been markedly improved and recommend acceptance of this manuscript.

Reviewer #2

(Remarks to the Author)

The authors have addressed all previous reviewer comments, including methodological clarifications, improved uncertainty quantification, figure revisions, and clearer interpretation of results. The revised manuscript presents a significant and timely contribution to the literature and is suitable for publication in its current form. Hence, I recommend acceptance.

Reviewer #3

(Remarks to the Author)

Reviewer #4

(Remarks to the Author)

I thank the authors for their thorough and thoughtful responses to my previous comments, as well as for the corresponding revisions made to the manuscript. The revisions have addressed all my concerns satisfactorily. The manuscript is now significantly improved, clearer, and well-supported. I have no further suggestions.

Response to Reviewers

1.0 Guidance for this document:

We thank the editorial team and the reviewers for their valuable comments and suggestions for our manuscript entitled “*National climate action can ameliorate, perpetuate, or exacerbate international air pollution inequalities*” that we submitted to *Nature Communications* earlier this year. We have endeavored to integrate these reviews and revise our manuscript to be less descriptive, to more directly connect the quantitative results to conclusions, and to link our findings to policy recommendations as discussed in our responses below. To differentiate comments from the reviewers, the authors, and new additions to the text please refer to the key:

Element	Text Style
Reviewer Comment	Italicized black text
Editor Comment	Italicized black text
Author Comment	Plain black text
Original Manuscript Text	Plain green text
Revised Manuscript Text	Plain blue text
Original Line Number	(in parentheses)
New Line Number	[in brackets]

2.0 NOTE TO ALL: In working through these revisions, we noticed an error in our original analysis in which health benefits from specific endpoints were accidentally excluded from our total estimates. This led to an underestimate of co-benefits in the original manuscript of ~56%. This mostly affected the total magnitude estimates (i.e., Table 1) and not the inequalities explored in the paper; however, there were some minor impacts on inequalities due to different baseline disease rates for different health outcomes across countries.

Additionally, given suggestions from the reviewers and editors, we have made major changes to aspects of this analysis. We have revised the original “imbalance ratio” and “exchange ratio” indicators into forms that are hopefully more intuitive and have explained these in detail in a new Figure (Figure 1), now included in the main text. Using these revised metrics, we have updated all of the figures and adjusted the language of the results to more quantitatively discuss inequalities and connect them to their causes and policy implications. While many of the ideas in the original manuscript are included here as well, we have endeavored to more clearly connect them to the quantitative results.

3.0 Reviewer Comments

3.1 Reviewer #1:

RIC0: *The manuscript evaluates how socioeconomic trends (SSPs) and climate mitigations (RCPs) might affect the health burdens associated with PM2.5 air pollution, and explores PM2.5 transboundary exchanges and associated environmental inequities across different climate scenarios. This manuscript is well organized and provides interesting results. However, several issues need to be addressed before the manuscript can be considered for publication.*

We thank Reviewer #1 for their positive comments; thank you for your suggested changes as we believe that these have greatly improved the quality of this work.

RIC1: *The study involves the coupling of multiple data and models, including GEOS-Chem, Health impact calculation, and so on. The authors should consider the uncertainty introduced by each link and its impact on the final results to increase the credibility.*

Thank you for raising this issue that we agree was missing in our original manuscript. To address this, we have now: (1) quantified uncertainty in our health impact calculation, (2) updated Table 1 and Figure 2 (previously Figure 1) to include uncertainty bounds (see section 4.0 of this document), (3) written a new uncertainty analysis section and added this to the methods, and (4) updated the discussion to go into more detail on these uncertainties. Beyond the health calculation, uncertainty is additionally introduced from the emission projections, chemical transport model simulation, and in the adjoint sensitivity calculation. In previous work¹ we found that the health impact calculation was the leading source of uncertainty; however, we recognize that this may not be the case when considering emission projections in the future. Please refer to our new updated “uncertainty analysis” section for our detailed thoughts on these uncertainties:

[743-782]

Uncertainty analysis

Assumptions and data that we use to estimate the health and equality impacts of climate action introduce uncertainty into our results and conclusions. Specifically, uncertainty is introduced primarily by: (1) emission projections, (2) the GEOS-Chem forward simulation, (3) the adjoint sensitivity calculation, and (4) the health impact analysis.

Although it is impossible to quantify a specific uncertainty associated with emissions projected for the future – given that the true values are unknown – the original dataset² was tested against historical emissions from another inventory, the Community Emissions Data System (CEDS)³ and the emissions were found to be well correlated for SO₂ from all sectors in 2005 ($R^2=0.71$). Additionally, they examined correlation with CEDS projections in 2050 for the energy sector and found moderate correlation for both SO₂ ($R^2=0.44$) and NO_x ($R^2=0.70$). Given the degree of uncertainty in these emission projections, we exclusively consider the relative differences in projections as opposed to their absolute implications for our analysis.

Although GEOS-Chem makes assumptions and simplifications to simulate the formation and transport of air pollution in a computationally tractable manner, it has a strong track record of

estimating atmospheric composition that compares well to observations^{4,5}. The version of the model that we use in this analysis does not include anthropogenic fugitive dust and secondary organic aerosol; the prior was estimated⁶ to increase global population-weighted PM_{2.5} by around 2.9 $\mu\text{g m}^{-3}$ and the latter was estimated⁷ to make up between 15%-30% of anthropogenic PM_{2.5} in urban environments. Fugitive dust PM_{2.5} is a primary species with a short atmospheric lifetime that mainly affects concentrations near where it is emitted. In contrast, secondary organic aerosol has a longer atmospheric lifetime enabling its transport across greater distances. Given the focus of our analysis on transboundary air pollution, it is likely that the exclusion of secondary organic aerosol from the simulation will lead to an underestimate of transboundary air pollution. Another source of uncertainty in our analysis is the application of local-linear sensitivities to inherently non-linear PM_{2.5} formation. Although the primary components of PM_{2.5} (i.e., black and organic carbonaceous aerosol) respond linearly to emissions, secondary inorganic aerosol does not respond linearly. Thus, this approach does not capture the second order effects of secondary inorganic aerosol.

Lastly, we quantify the uncertainty introduced from the health impact analysis by using upper and lower bound estimates of the different components of the health calculation (i.e., the relative risk, population estimates, and baseline disease rates) using the lower and upper bound values provided by the GBD study. This uncertainty is the only value included in the uncertainty bounds presented throughout the text given that previous work suggests that this is the largest contributor to uncertainty in our approach¹. We do not consider uncertainty in the projected changes to health data (i.e., the projected population and disease rates for 2040).

And refer to our updates to the discussion section on uncertainty:

(384-402)

There are many factors that could influence the unequal distribution of air pollution across different socioeconomic trends and mitigation strategies beyond what is considered in this study. Specifically, our adjoint-derived methodology calculates linear relationships between emissions and PM_{2.5} and while some components of PM_{2.5} (e.g., primary carbonaceous aerosol) are formed linearly, others (e.g., secondary inorganic aerosol) are formed through complex non-linear chemistry⁸. Our analysis relies on the simplified assumption that the pollution formation relationships calculated in our simulation will be maintained in the future, although we expect changes to climate^{9,10} and chemical^{11,12} environment would be additional modifiers to consider. Additionally, our analysis neglects to include secondary organic aerosol formation that makes up a substantial component of total PM_{2.5}, especially in heavily populated urban areas¹³. Given the longer atmospheric lifetime of this component, it is likely that we underestimate some of the long-range exchange of air pollution. Additionally, this analysis is conducted for emissions scenarios in a single year – 2040 – so it represents a snapshot in time of the differences across these socioeconomic trends and mitigation strategies. Given that the SSP-RCP emissions are designed to peak at different points across different countries and regions, it is likely that these imbalances and exchanges will differ at other points in the future. While in this study we do not explore this in detail, a benefit of this adjoint methodology is its ability to rapidly assess different years – as well as different scenarios – offline without additional simulations. To demonstrate this, we include figures for an additional year (i.e., 2030) in the supplement (Figure S2-S5).

[530-553]

The results presented in our work should be considered alongside sources of uncertainty. First, our adjoint-derived methodology calculates linear relationships between emissions and PM_{2.5} and while some components of PM_{2.5} (e.g., primary carbonaceous aerosol) are formed linearly, others (e.g., secondary inorganic aerosol) are formed through complex non-linear chemistry⁸. Additionally, our estimates of co-benefit contributions rely on the simplified assumption that the pollution formation relationships that we calculate for our base year (i.e., 2010) will be maintained in the future. We anticipate that changes to climate^{9,10} and chemical^{11,12} environment in the future will modify transboundary air pollution and we suggest that feedbacks between climate change and transboundary air pollution are explored in future work. Our analysis neglects to include anthropogenic fugitive dust and secondary organic aerosol formation that contribute to PM_{2.5}, especially in heavily populated urban areas⁷. Given the longer atmospheric lifetime of secondary organic aerosol, it is likely that we underestimate some of the long-range exchange of air pollution. Additionally, this analysis is conducted for emissions scenarios in a single year – 2040 – so it represents a snapshot in time of the differences between these socioeconomic trends and mitigation strategies. Given that the SSP-RCP emissions are designed to peak at different points across different countries and regions, it is likely that these transboundary fractions and exchanges will differ at other points in the future. While in this study we do not explore this in detail, a benefit of this adjoint methodology is its ability to rapidly assess different years – as well as different scenarios – offline without additional simulations. To demonstrate this, we include figures for an additional year (i.e., 2030) in the supplement (Figure S6-S9). Lastly, although we estimate uncertainty associated with the health calculation, there is additional uncertainty in the projected changes to population, age distribution, and baseline disease rates that are not captured in our analysis along with uncertainty on the air quality modeling side as we discuss in the methods.

R1C2: *The PM_{2.5} transboundary exchanges include atmospheric transport and trade exchange, as should be noted in the text. And there are some studies that have revealed the impact of atmospheric transport on health and environmental inequities in historical years that the authors can compare the findings of this paper with them.*

We agree with the reviewer that transboundary exchanges of PM_{2.5} include both an atmospheric transport component and exchanges associated with trade (i.e., production and consumption relationships). We have now clarified this in the text and cited some recent relevant studies:

[63-64]

Atmospheric transport drives some of this exchange; however, trade imbalances in consumption and production contribute to air pollution inequalities as well^{14,15}.

Additionally, as you suggest we have now compared our analysis to some prior work. We note that in our analysis we find a higher transboundary component than some past work; however, this is likely due to the greater granularity (i.e., countries instead of regions) that our adjoint analysis enables as we discuss in the text:

[269-280]

We note that these transboundary contributions are substantially greater than past analyses^{16,17} of historical transboundary air pollution burdens, which found 14% and 12% of all PM_{2.5}-related deaths were attributable to transboundary pollution. This difference is likely attributable to three main factors: (1) these past studies focused on aggregated groups of countries whose boundaries are further from peak emissions locations whereas we perform this calculation for individual countries, (2) in this study we consider climate action whereas these previous studies consider all anthropogenic emissions; climate action is not adopted to the same degree globally and many of the highest emission reductions are implemented in a small set of developed nations, and (3) many of the targets of climate action are precursors of secondary PM_{2.5} (specifically NO_x and SO₂) and not primary PM_{2.5} and thus this action targets pollution that is more likely to be transported greater distances due to its longer atmospheric lifetime.

RIC3: *The manuscript defines two metrics to describe unfairness, the imbalance ratio and the exchange ratio, but their descriptions in the main text are too difficult to understand, such as Line538-555, Line54-56. The authors' intent is not understood until Figure S1.*

Our original descriptions of the imbalance and exchange ratios were lacking and were not placed in an appropriate area as you suggest (other reviewers have commented on this as well). To address this, we have made major modifications. First, we have reworked the original indicators into quantities that are hopefully more intuitive, and we have developed a new diagram explaining the terms (originally Figure S1 now Figure 1) and moved this to the beginning of the results section so that readers encounter it immediately (see section 4.0 of this document for the new figure). Lastly, we have enhanced descriptions of the metrics throughout:

(115-129)

To further explore how socioeconomic trends and mitigation strategies vary across different geographies, we group all countries into one of six regions: South America (SA), Oceania (OC), North America (NA), Europe (EU), Asia (AS), and Africa (AF) (Figure 1). The specific countries in each of these regions are provided in the supplement (Table S1). We estimate the region-specific co-benefits (i.e., deaths avoided from climate action) across all scenarios (Figure 1a) and these same co-benefits normalized for each scenario (Figure 1b). Scenarios in which climate action is less aggressive – such as trends towards inequality and fragmentation and higher radiative forcing (i.e., higher RCPs) – lead to fewer health benefits from climate action. This is consistent across the six regions with a majority of the health benefits occurring in Asian countries, followed by Europe, Africa, North America, and then South America. In sustainable futures, such as SSP1-19, co-benefits in Asian countries make up the largest percentage of the global co-benefits (72%), more so than Europe (13%), Africa (8%), North America (4%) and South America (3%). However, in fragmented futures, such as SSP3-60, the Asian share of co-benefits grows (83%) and the share of other regions decreases relatively for Europe (8%), Africa (5%), North America (3%), and South America (1%).

[148-160]

Past studies on the impact of climate action on air quality generally estimate the total magnitude of health benefits (as we do in the previous paragraphs); however, this neglects a crucial aspect of climate action with implications for equality: transboundary air pollution. To characterize disparities induced by changes to transboundary air pollution we define two metrics that are

associated with air pollution inequality: “transboundary fractions” and “exchanges” (Figure 1). When global climate action is adopted, a fraction of the co-benefits accrued by a country are attributable to external action (i.e., climate mitigation outside of its borders). We define a “transboundary fraction” (Figure 1a) that quantifies the fraction of all co-benefits in a country that originate from action outside that country (refer to the methods section for more details). The fraction can also be calculated for regions of aggregated countries. Higher transboundary fractions indicate a country or region that requires more regional or global cooperation to realize the benefits of climate action whereas a lower transboundary fraction positions a country or region as largely being in control of their own climate co-benefits.

(237-244)

The benefits associated with climate action in one country are not exclusively realized within the same country. To unpack how intraregional and interregional exchanges in climate action benefits could differ across different socioeconomic trends and mitigation strategies, we calculate “exchange ratios” that represent the transboundary exchange of air pollution health impacts from one region to another. The exchange ratio is defined as the ratio of the benefits occurring in one country owing to emissions reductions in a second country, divided by the benefits in the second country owing to emissions reductions in the first country, as shown in more detail in the methods and schematically in the supplement (Figure S1b).

[326-335]

The exchange of climate co-benefits between pairs of countries (or regions) evolves in response to different socioeconomic development pathways and climate mitigation strategies. We characterize the intraregional and interregional exchange of climate co-benefits by calculating “exchanges” (Figure 1b) that represent the air pollution co-benefits realized in one country that are attributable to climate action in another country. In our analyses, we consider exchanges in three ways: (1) in their absolute sense an “exchange” represents how emission reduction in one country induces co-benefits in another, (2) as “net exchanges” by differencing the two exchanges in a pair, and (3) the proportion of co-benefits contributed by one member of a pair to the total gross co-benefits exchanged between the pair to determine if a specific country dominates the exchange of air pollution co-benefits.

[710-723]

To explore how inequalities in transboundary air pollution – and the associated co-benefits from climate emission scenarios – vary across different climate futures, we define a set of metrics. First, we consider the transboundary fraction:

$$TF_{m,s} = \frac{\sum_{I \notin m} \Delta J_{I,m,s}}{\sum_I \Delta J_{I,m,s}} \quad (5)$$

The transboundary fraction ($TF_{m,s}$) for country m and scenario s is equivalent to the ratio of the co-benefits contributed to country m from emissions outside of the country compared to the co-benefits contributed from everywhere, including the country. This is calculated by considering emission reductions that occur in grid cells that are not within m ($I \notin m$), over the benefits in country m contributed by emission reductions from all grid cells. A transboundary fraction close to one indicates that country m receives most of its benefits through external emission

reductions, whereas a ratio close to zero indicates that domestic emission reductions contribute to more benefits within m than reductions elsewhere.

(546-551)

The exchange ratio ($ER_{m,n,s}$) between country m and n for scenario s is equivalent to the ratio between the benefits in m from emission reductions in n over the benefits in n from emission reductions in m . An exchange ratio of 1 indicates that the two countries contribute equally to benefits from climate action, a value greater than 1 indicates that m receives more benefits from n than vice versa, and a value less than 1 indicates that n receives more benefits from m than vice versa. Schematic depictions of both ratios are included in the supplement (Figure S1).

[723-741]

Next, we consider exchanges:

$$EXC_{m,n,s} = \quad (6)$$

The exchange ($EXC_{m,n,s}$) between countries or regions m and n for scenario s is equivalent to the amount of air pollution-related benefits in n that are attributable to emission reductions in m for scenario s . The “net exchange” refers to the difference between two exchanges in a country pair (i.e., $EXC_{m,n,s} - EXC_{n,m,s}$). Lastly, the “contribution to total co-benefits exchanged”, used in Figure 5, refers to the percentage of total co-benefits exchanged between a pair attributable to one member of the pair:

$$TEC_{m,n,s} = \frac{EXC_{m,n,s}}{EXC_{m,n,s} + EXC_{n,m,s}} \quad (7)$$

Thus, the contribution to total co-benefits exchanged, $TEC_{m,n,s}$, compares the exchange contributed by country m to country n ($EXC_{m,n,s}$) to the total gross co-benefits exchanged between the two countries ($EXC_{m,n,s} + EXC_{n,m,s}$). A TEC value of 0.5 indicates balanced contributions to co-benefits exchanged between countries, while values above 0.5 indicate greater contributions from country m and values below 0.5 indicate greater contributions from country n . Schematic depictions of these metrics are included Figure 1.

R1C4: *Based on the impacts of climate action on PM2.5-related health and inequity analyzed in this paper, the authors can make specific recommendations for climate pathways or mitigation actions in future years.*

We agree that there was a need in our original manuscript to better connect our results to specific policy recommendations; this has also been raised by other reviewers and the editorial team. We have now made major revisions throughout the text to address this and have dedicated a specific paragraph in the discussion for this purpose. Please refer to our responses to Reviewer #2 (R2C2) and (R2C5) and Reviewer #4 (R4C8) for details.

R1C5: *What is the basis for the article's choice of 2040 as the study year? The results for 2030 are also presented in SI; is there any difference in the findings on inequality between the two years?*

Thank you for raising this issue as it was not discussed in the original manuscript, we choose the year 2040 for three reasons: (1) it is the earliest year in which most of the SSP3 scenarios diverge from SSP3-Baseline, (2) it is the latest year for which we have GBD Foresights projections, and (3) it is approximately when global NO_x emissions peak. This was not mentioned in the original manuscript and so we have now updated the text to include this explanation:

[670-673]

We note that we selected 2040 as the main year for our analysis because it was the first year in which SSP3 scenarios began to deviate from the SSP3-Baseline, it was the latest year for which we had GBD Foresights data (see next section), and approximately when global NO_x emissions peak for the SSP3-Baseline scenario.

To respond to your other comment regarding differences in the findings on inequality between the two years, we agree that we do not give much detail on the differences in inequality between 2030 and 2040. This was intentional, as quantifying and interpreting interannual changes in co-benefits and inequality is a large undertaking and likely warrants an additional follow-up study; however, we have added some more detail to this by creating a new supplement Figure (Figure S2) that compares imbalance ratios between 2040 and 2030 in the style of Figure 3 (see section 4.0) and have discussed it in the text:

[303-308]

Although we focus primarily on 2040 in our analysis, we additionally calculate the differences in transboundary fractions between the base year of study (2040) and 2030 (Figure S2) across SSP1-26 and SSP3-26. We find that in 2040 transboundary fractions are moderately higher in China, India, the US, and Brazil and lower in parts of Africa compared to SSP1 in 2030; however, these interannual changes are smaller than those attributable to socioeconomic changes.

Additionally, we refer to this as a potential source of uncertainty to consider in the discussion (see our response to R1C1).

R1C6: *Confidence intervals are missing for some of the premature deaths.*

Thank you for pointing this out, this has now been resolved.

R1C7: *What are the data sources for baseline incidence and population in 2040? national scale?*

You are correct, this has now been mentioned explicitly in the text:

(508-510)

We combine these relative risks with baseline disease rates and population data from the GBD 2019 study for a base year of 2010 – the same as the adjoint sensitivities.

[687-689]

We combine these relative risks with national baseline disease rates and population data from the GBD 2019 study for a base year of 2010 – the same as the adjoint sensitivities.

R1C8: *The font size of Figure 1 (a) (b) and Figure 3 (a) is too small. Consider enlarging or using abbreviations.*

Thank you for this suggestion, we have now made major changes to Figure 2 (formerly Figure 1) and Figure 4 (formerly Figure 3) to improve readability (see section 4.0 of this document). We have also enhanced the image resolution to retain more detail in the figures when zooming in.

3.2. Reviewer #2:

R2C0: *While the manuscript presents a comprehensive assessment of transboundary PM2.5-related health co-benefits under various climate futures, and introduces metrics such as the imbalance ratio and exchange ratio, the overall methodological innovation is incremental rather than groundbreaking. The study builds upon previously published adjoint-based sensitivity analyses, extending the scale and scope, but without introducing novel modeling frameworks or substantially new insights.*

We thank Reviewer #2 for their helpful comments and appreciate their insights. We believe that through incorporating your suggestions this manuscript has greatly improved. We address the specific criticism on novelty in response to your next comment.

R2C1: *The methodology is largely based on previous work. The modeling approach, including the use of the GEOS-Chem adjoint model for estimating sensitivities of PM2.5 exposure to precursor emissions, has been described in several earlier studies. The integration with health impact assessment methods and SSP-RCP emissions scenarios also follows a structure that has previously been applied. In the current manuscript, the methodology is scaled to a global level and applied to a broader set of scenarios. However, no substantial changes to the model, workflow, or conceptual framework are introduced. As such, the novelty of the methodology appears to be limited. For a journal that prioritizes conceptual or methodological advancement, this may not be sufficient.*

Thank you for raising this important point for discussion. We apologize that in the original version of the manuscript the novelty of this work was not clearly stated. Through updating the figures and reworking large sections of the results and discussion based on suggestions from yourself and other reviewers we believe that the novelty of this work is now clearer and more well-articulated. We believe that there are many advancements in our analysis that positions it as an appropriate entry into the *Justice and Equality in Decarbonization* collection in *Nature Communications* as we elaborate on below.

While the underlying modeling approach that we have used here has indeed been described in our prior work, this is the first time that any detailed air quality modeling study has developed or used a source-receptor matrix for almost every individual country in the world. Regardless of the technical approach, we know of no prior work to achieve this, and thus the results here are technically and scientifically quite novel in that we resolve the long-range transport between almost every pair of countries. Our approach enables probing of policy-related questions throughout the globe without selection or aggregation biases, in contrast to previous studies that have focused on a subset of countries or aggregations of countries, often to semi-continental or continental scales. Although recent studies^{18,19} have developed means of connecting air quality and climate effects of emissions controls to national impacts; these studies do not explore geographic variation in both the policy implementation and health impacts as we do here. We find, for example, a more significant role of atmospheric transport than prior studies^{16,20} owing to our ability to resolve individual countries. Further, for this study alone we include adjoint sensitivities calculated for 125 more countries than our previous study²¹ that focused just on the G20. This means that ~75% of these adjoint-based sensitivities are unique to this study and have

never been used in a scientific study before. Crucially, our prior analysis focused primarily on the Global North (i.e., G20); however, there is a well-documented understudy bias of the Global South, and the results of our approach enable previously impossible analyses of inequalities in less developed nations as can be seen through many of our results and discussion that are focused on African countries.

Second, we agree that fundamentals of our method are built upon prior work; however, we would argue that this is a positive aspect of our analysis. These are now established methods being applied to novel questions that have never been examined with this degree of detail but, crucially, benefit from the comprehensive source-receptor specificity that this new set of adjoint sensitivities enable. This allow us to resolve previously undocumented details regarding inequality associated with climate action.

Lastly, we would note that none of the reviewers have compared our inequality analyses to other studies, as no others exist -- further supporting the novelty of this work. We now explicitly call to attention the novelty of this work in the introduction:

(86-88)

We identify not only the health impacts from specific scenarios but also – by leveraging the adjoint model source-receptor relationships – identify how each country contributes to impacts throughout the world.

[93-99]

These adjoint modeling results enable us to explore the impacts of climate action across a larger geographic extent (only 25% of these sensitivities have been presented in prior analysis²²), allow us to uncover novel information about climate action in the Global South that has historically been understudied, and, crucially, represent the first development of a source-receptor matrix for nearly every individual country facilitating the investigation of policy-related questions throughout the globe without selection or aggregation biases.

We also have rewritten parts of the discussion text to emphasize this:

(339-348)

Through the application of adjoint sensitivities in conjunction with SSP-RCP emission projections, we characterize the benefits of climate action and consider how the exchange and imbalance relationships between countries and regions could evolve across different scenarios. The approach developed in this work introduces methods to characterize how source receptor relationships change in response to different challenges to mitigation and adaptation and offers a means to investigate how climate action could affect inequalities in the global distribution and exchange of air pollution. This progresses beyond prior analyses that focused on quantifying the magnitude of benefits from emission reductions by investigating the complex exchange of air pollution and health impacts between countries and regions.

[457-465]

We integrate adjoint sensitivities with SSP and RCP emission projection to extend beyond past work that estimated total co-benefits from climate action and consider how transboundary co-benefits and exchange relationships between countries and regions could change across different climate scenarios in 2040. We introduce novel capabilities for characterizing how country-specific source receptor relationships worldwide change in response to different challenges to mitigation and adaptation. This enables us to investigate how climate action could affect inequalities in the global distribution and exchange of air pollution that progresses beyond prior analyses that focused on quantifying the magnitude of benefits from emission reductions or examined aggregated¹⁷ or limited²² source receptor relationships.

R2C2: *While the imbalance and exchange ratios are introduced as new indicators, they are derived directly from existing model outputs and are relatively straightforward in formulation. Their interpretation remains largely descriptive, and their implications are not developed in depth. The study shows that some regions receive more co-benefits than they contribute and that these patterns vary across climate scenarios. However, these observations tend to confirm existing assumptions about global inequalities rather than reveal new or unexpected findings. It is not shown how these indicators would be useful for designing future policy or guiding negotiations on climate action and air quality management.*

Based on feedback from yourself and other reviewers it became clear that the old indicators (i.e., imbalance and exchange ratios) were problematic and our interpretations of them could be improved through more quantitative discussion that explored the implications of their values. To address this, we have reworked the old indicators into a new set of simpler indicators (see also our response to R1C3) and now describe these in detail at the beginning of the results section along with presenting them schematically and discussing their implications (see Figure 1):

(133-141)

We define an “imbalance ratio” that compares the global benefits (deaths avoided) induced by climate mitigation in a country compared to the benefits realized in that country from action everywhere (Figure 1c). The imbalance ratio is defined in the methods and explained schematically in the supplement (Figure S1a). We exclude the imbalance ratio from countries in Oceania as they are much higher than the other five regions given their substantially smaller populations. The regions fall into one of three categories: their countries contribute more benefits than they receive regardless of scenario (i.e., imbalances greater than one), their countries contribute less benefits than they receive regardless of scenario (i.e., imbalances less than one), or they have scenario-dependent imbalances (i.e., imbalances are both greater and less than one).

[148-160]

Past studies on the impact of climate action on air quality generally estimate the total magnitude of health benefits (as we do in the previous paragraphs); however, this neglects a crucial aspect of climate action with implications for equality: transboundary air pollution. To characterize disparities induced by changes to transboundary air pollution we define two metrics that are associated with air pollution inequality: “transboundary fractions” and “exchanges” (Figure 1). When global climate action is adopted, a fraction of the co-benefits accrued by a country are attributable to external action (i.e., climate mitigation outside of its borders). We define a “transboundary fraction” (Figure 1a) that quantifies the fraction of all co-benefits in a country

that originate from action outside that country (refer to the methods section for more details). The fraction can also be calculated for regions of aggregated countries. Higher transboundary fractions indicate a country or region that requires more regional or global cooperation to realize the benefits of climate action whereas a lower transboundary fraction positions a country or region as largely being in control of their own climate co-benefits.

(238-244)

To unpack how intraregional and interregional exchanges in climate action benefits could differ across different socioeconomic trends and mitigation strategies, we calculate “exchange ratios” that represent the transboundary exchange of air pollution health impacts from one region to another. The exchange ratio is defined as the ratio of the benefits occurring in one country owing to emissions reductions in a second country, divided by the benefits in the second country owing to emissions reductions in the first country, as shown in more detail in the methods and schematically in the supplement (Figure S1b).

[326-335]

The exchange of climate co-benefits between pairs of countries (or regions) evolves in response to different socioeconomic development pathways and climate mitigation strategies. We characterize the intraregional and interregional exchange of climate co-benefits by calculating “exchanges” (Figure 1b) that represent the air pollution co-benefits realized in one country that are attributable to climate action in another country. In our analyses, we consider exchanges in three ways: (1) in their absolute sense an “exchange” represents how emission reduction in one country induces co-benefits in another, (2) as “net exchanges” by differencing the two exchanges in a pair, and (3) the proportion of co-benefits contributed by one member of a pair to the total gross co-benefits exchanged between the pair to determine if a specific country dominates the exchange of air pollution co-benefits.

Second, we agree that our interpretations of the old indicators were mostly descriptive, and that we did not fully develop their implications. To address this, we have now discussed these indicators more quantitatively and further expand upon the implications of different values. We have done this throughout the results text but highlight a few examples below:

[209-218]

Africa is the region most affected by regionally external climate action: on average 12% (9%, 19%) of its co-benefits originate from action in another region. This suggests that African countries are especially reliant on external action to maximize their climate co-benefits. This reliance is mediated by the degree of mitigation action. From RCP-45 to RCP-26, stronger mitigation induces proportionally more transboundary co-benefits across the different socioeconomic development pathways ranging from 8% more in SSP1 to 53% more in SSP3. This relationship between climate mitigation and transboundary co-benefits in Africa is likely attributable to development. The most aggressive climate mitigation scenarios necessitate stronger action in countries that have historically polluted more (e.g., countries in Europe) and so, as these countries adopt stronger climate action, Africa benefits proportionally more.

[251-261]

Transboundary fractions are more highly variable for countries than regions: e.g., there is a standard deviation of 0.29 for countries compared to the 0.04 for regions in SSP1-26. This variability in the transboundary fraction is influenced by the degree of development (Figure 3). For SSP1-26, the transboundary fraction for the twenty countries with the lowest Human Development Index (HDI)²³ is on average 0.76 compared to 0.65 for the twenty highest HDI countries. Additionally, 90% of these low HDI countries are African nations with the two exceptions being Afghanistan (0.91) and Yemen (0.78). In contrast, the twenty countries with the highest HDI are primarily from Europe, North America, Oceania, and Asia; none of these countries are from Africa or South America. This suggests that developed nations receive proportionally fewer of their co-benefits from external action than developing nations and subsequently that global cooperation benefits developing nations more than developed ones.

[352-367]

These exchange inequalities can be exacerbated or ameliorated through different socioeconomic development pathways. For Africa, fragmented socioeconomic development transforms the relatively equivalent exchange between South Africa and Mozambique in SSP1-26 to one in which South Africa contributes much more (3.39 times). For Asia, the previously equivalent Pakistan-India exchange in SSP1-26 has greater contributions from India (1.33) in SSP3-26. The inequal exchange between China and India is perpetuated in middle-of-the-road (1.79), fragmented (1.76), and inequality (1.82) development pathways; however, it notably increases through conventional development (2.07). Lastly, in Europe, fragmented development exacerbates the exchange inequality between Spain and Portugal (5.78) but partially reduces the inequality between France and Spain (3.67). Many of the European co-benefit exchanges increase in conventional development; this is likely owing to higher climate mitigation (RCP-26) paired with conventional development (SSP5) which necessitates greater climate action in developed countries (including many in Europe) compared to other socioeconomic pathways in which there is more even sustainable development. When considering weaker climate mitigation as we include in the supplement (Figures S3 and S4), we see that many of these increased exchanges are resolved.

[411-424]

There is even greater sensitivity to socioeconomic development at the country-scale (Figure 5b). When transitioning from fragmented (SSP3-45) to sustainable (SSP1-45) development there is generally more balance between country exchange pairs especially in the pairs that undergo the largest change to exchanges: in the top ten pairs, the country with the higher GDP at present contributed 61% of co-benefits in SSP3-45 compared to 50% in SSP1-45. Nine of these ten pairs had more balanced exchange patterns (i.e., closer to 50%) in sustainable development futures compared to fragmentation futures. Crucially, sustainable development consistently pushed towards greater contributions from the lower GDP member of the exchange pair, ranging from 32% lower in the India-Nepal pair to 19% lower in Spain-Morocco. This supports previous findings that more sustainable futures will enable greater contributions to co-benefits from poorer nations due to greater development and industrialization. In the transition instead from SSP2-45 to SSP1-45 (Figure 5d), this trend towards more balance does not occur; average contributions from the higher GDP change only from 57% in SSP2-45 to 54% in SSP1-45 in the top 10 nations.

Third, we also agree that our original manuscript did not do a good enough job at highlighting unexpected or surprising results. In this revision we have taken care to specifically call out surprising or unexpected results and include a few examples below:

[194-205]

Interestingly, although Asia makes up a dominant share of the total co-benefits from climate action, the same is not true for transboundary co-benefits (i.e., co-benefits originating from emissions outside of the region that they benefit) (Figure 2b). In SSP1-19, Asia has the largest share of transboundary co-benefits (38%); however, this is only slightly ahead of other regions such as Europe (32%) and Africa (21%). In some scenarios, such as SSP1-45, Asia actually makes up a smaller percentage of transboundary co-benefits (32%) than Europe (36%) and only slightly more than Africa (22%) despite its much larger population. This is an unexpected result. As mentioned previously, rapid development and a large aging population position Asia as the primary benefactor of climate action; however, this is not the case for transboundary co-benefits. Crucially, the major polluters and population centers of Asia in China and India are located in the eastern part of the continent away from other regions; thus, a majority (62% in SSP4-45) of the benefits of climate action in the continent are geographically isolated from other regions.

[269-280]

We note that these transboundary contributions are substantially greater than past analyses^{16,17} of historical transboundary air pollution burdens, which found 14% and 12% of all PM_{2.5}-related deaths were attributable to transboundary pollution. This difference is likely attributable to three main factors: (1) these past studies focused on aggregated groups of countries whose boundaries are further from peak emissions locations whereas we perform this calculation for individual countries, (2) in this study we consider climate action whereas these previous studies consider all anthropogenic emissions; climate action is not adopted to the same degree globally and many of the highest emission reductions are implemented in a small set of developed nations, and (3) many of the targets of climate action are precursors of secondary PM_{2.5} (specifically NO_x and SO₂) and not primary PM_{2.5} and thus this action targets pollution that is more likely to be transported greater distances due to its longer atmospheric lifetime.

[227-238]

Surprisingly, less equal socioeconomic development increases the transboundary fraction for Africa. One reason why this could occur is that in more fragmented climate futures, there is less cooperative development. Subsequently, in these less equal scenarios, African countries become even more sensitive to global action (67% higher in SSP3-26 than SSP1-26) as there is less local development and industrialization. This relationship is true across all of the RCPs; however, it is stronger in those with higher climate mitigation (i.e., RCP-26) than those with weaker mitigation. Specifically, from SSP3-45 to SSP1-45, the transboundary fraction only increases by 18%. Ultimately the transboundary fraction in Africa is mediated by two competing phenomena: (1) more aggressive climate mitigation necessitates proportionally more involvement from developed nations and thus higher transboundary co-benefits and (2) less equal socioeconomic development further increases the disparity in industrialization and development in many African countries which forces them to be more dependent on external action to maximize co-benefits.

Lastly, we have now dedicated text specifically discussing how these indicators could be useful for policy design in the discussion:

[555-561]

With these uncertainties in mind, our results have clear relevance for climate policy design. First, the specific inequality metrics we define, transboundary fractions and exchanges, have implications for policy. In quantifying the fraction of co-benefits that are transboundary, countries and groups can identify the scenarios that are most beneficial in specifically alleviating transboundary air pollution external to their borders. Through quantifying exchanges between countries or regions, they can identify the most important partnerships to address imbalanced exchanges of air pollution.

R2C3: *All results are presented for the year 2040. Since SSP-RCP pathways evolve differently over time, it is unclear whether the identified patterns of imbalance and exchange are consistent across other time points. A single-year analysis may be influenced by scenario-specific assumptions in that year and does not allow for understanding temporal trends or cumulative effects. Although additional figures for 2030 are provided in the supplement, the main analysis remains focused on one year, which limits the robustness of the conclusions.*

We agree that the patterns of imbalance and exchange will vary through time. First, we have now created a new figure (Figure S2) that directly compares 2030 and 2040 transboundary fractions and comment on this in the results. For this scenario we see that different years can influence transboundary air pollution; however, generally this is a smaller effect than what we see from distinct socioeconomic development.

[303-312]

Although we focus primarily on 2040 in our analysis, we additionally calculate the differences in transboundary fractions between the base year of study (2040) and 2030 (Figure S2) across SSP1-26 and SSP3-26. We find that in 2040 transboundary fractions are moderately higher in China, India, the US, and Brazil and lower in parts of Africa compared to SSP1 in 2030; however, these interannual changes are smaller than those attributable to socioeconomic changes. Ultimately, these findings imply two key outcomes: (1) the effects of climate action on transboundary air pollution are unique to socioeconomic development and mitigation strategies and these effects are not additive and (2) less developed countries receive greater transboundary co-benefits in less equal socioeconomic development as they are more dependent on external action due to weaker internal development and industrialization.

That being said, we agree that we need to make it clear that these results are most relevant for 2040 and could deviate for different years in our analysis. To do this we have updated some text in the introduction and discussion:

(86-88)

We identify not only the health impacts from specific scenarios but also – by leveraging the adjoint model source-receptor relationships – identify how each country contributes to impacts throughout the world.

[88-90]

We identify not only the health impacts from specific scenarios but also – by leveraging the adjoint model source-receptor relationships – identify how each country contributes to impacts throughout the world in 2040.

[457-460]

We integrate adjoint sensitivities with SSP and RCP emission projection to extend beyond past work that estimated total co-benefits from climate action and consider how transboundary co-benefits and exchange relationships between countries and regions could change across different climate scenarios in 2040.

We also note that this was discussed briefly in the original manuscript and have kept this text in our uncertainty paragraph in the discussion:

(395-399)

Additionally, this analysis is conducted for emissions scenarios in a single year – 2040 – so it represents a snapshot in time of the differences across these socioeconomic trends and mitigation strategies. Given that the SSP-RCP emissions are designed to peak at different points across different countries and regions, it is likely that these imbalances and exchanges will differ at other points in the future.

R2C4: *The version of GEOS-Chem used in this study does not include secondary organic aerosol formation, which can contribute significantly to PM_{2.5}, especially in urban and vegetated regions. The lack of these components may lead to an underestimation of long-range transport and influence the imbalance and exchange results. Additionally, the adjoint sensitivities are based on linear approximations, which may not fully capture the nonlinear chemical reactions involved in PM_{2.5} formation, particularly in areas with high emissions of ammonia and nitrogen oxides. The limitations of the chemical representation are acknowledged briefly in the methods section, but their implications for the main findings are not discussed in detail.*

Thank you for this comment and we agree that we could discuss the implications of these uncertainties in greater detail. We have now addressed this by developing an uncertainty analysis (see our response to R1C1). Additionally, we have adjusted some of the text in the discussion on this point to better highlight these limitations:

(389-395)

Our analysis relies on the simplified assumption that the pollution formation relationships calculated in our simulation will be maintained in the future, although we expect changes to climate^{9,10} and chemical^{11,12} environment would be additional modifiers to consider. Additionally, our analysis neglects to include secondary organic aerosol formation that makes up a substantial component of total PM_{2.5}, especially in heavily populated urban areas¹³. Given the longer atmospheric lifetime of this component, it is likely that we underestimate some of the long-range exchange of air pollution.

[534-542]

Additionally, our estimates of co-benefit contributions rely on the simplified assumption that the pollution formation relationships that we calculate for our base year (i.e., 2010) will be maintained in the future. We anticipate that changes to climate^{9,10} and chemical^{11,12} environment in the future will modify transboundary air pollution and we suggest that feedbacks between climate change and transboundary air pollution are explored in future work. Our analysis neglects to include anthropogenic fugitive dust and secondary organic aerosol formation that contribute to PM_{2.5}, especially in heavily populated urban areas⁷. Given the longer atmospheric lifetime of secondary organic aerosol, it is likely that we underestimate some of the long-range exchange of air pollution.

R2C5: *Although the manuscript touches on environmental justice and potential inequalities in the distribution of air pollution benefits, it is not made clear how the presented metrics could be applied in practice. There is a lack of connection between the model outputs and current international climate mechanisms or policy instruments. The imbalance ratio, in particular, is discussed without a clear normative interpretation. It remains uncertain whether a high or low ratio should be viewed as desirable or problematic, and how such findings could inform equitable burden sharing in climate action.*

Thank you for raising this point. We have mostly responded to this concern in response to your second comment (R2C2). Beyond what we include there we have also now made it a point to better connect our outcomes to policy implications; for this please refer to our responses to R4C8 and some additional examples below:

[186-192]

This high share of co-benefits in Asia is not surprising and is attributable to two factors: (1) the population of Asia is large at present and by 2040 this young population will age and become more susceptible to PM_{2.5}-related health impacts and (2) China and India are developing rapidly and these two countries receive a majority of the PM_{2.5}-related deaths in the region, thus climate action in these countries could alleviate a great degree of the air pollution health burden. Specifically, co-benefits in these two countries alone make up between 52% (SSP1-19) and 63% (SSP3-60) of all global co-benefits.

[490-499]

For both regions and countries, the fraction of co-benefits that are transboundary is closely tied to the scale of climate mitigation and the socioeconomic development pathway in which the action takes place. African countries exhibit higher transboundary fractions in high mitigation scenarios and less equal socioeconomic development. The former relationship occurs because high mitigation scenarios require more proportionate action from more economically developed areas. The latter relationship is due to the fact that more fragmented socioeconomic development induces greater development disparities and subsequently more global dependence on external action throughout many countries in Africa and other developing areas of the world. This implies a need for nuanced climate policy design that incorporates climate mitigation action in tandem with projected socioeconomic development to ensure more equitable climate co-benefits.

[561-575]

Second, there is clear evidence from our results that socioeconomic development and mitigation action are best considered in parallel rather than separately; the socioeconomic environment in which mitigation occurs dramatically influences both the total magnitude of co-benefits and the inequalities associated with transboundary air pollution. More fragmented socioeconomic development increases transboundary air pollution in developing countries (especially in Africa) and makes them more beholden to the decision-making of developed countries. Stronger climate mitigation, although overall beneficial, also increases the fraction of co-benefits that are transboundary as the most aggressive climate action requires greater buy-in from historically polluting developed nations. From a socioeconomic development perspective, transitioning from fragmented to sustainable climate futures leads to more balanced exchanges of co-benefits which enables developing countries to contribute proportionally more to exchanges due to greater industrialization and development. Ultimately, this suggests that sustainable socioeconomic development enables developing countries to participate more in global climate action, thus benefitting both themselves and their wealthier and more developed neighbors via improved domestic and foreign air quality.

R2C6: *In lines 99–101, the main text states that SSP1 avoids 0.53 million deaths, but Table 1 reports 581,000 (i.e., 0.581 million) deaths for SSP1-19. While the values are close, the correspondence between the text and specific scenarios in Table 1 should be clarified to avoid confusion.*

Thank you for pointing this out. In the original manuscript we had calculated the average across all RCPs, but this was not mentioned in the text. We have now just included a single SSP-RCP combination for each of the different socioeconomic development pathways:

(97-109)

The SSP-RCP scenarios – and the socioeconomic trends and mitigation strategies they represent – project lower emissions in 2040 than the worst-case scenario (i.e., SSP3-Baseline) that result in improved air quality and reduced health impacts; however, emission reduction impacts vary by scenario (Table 1). For example, socioeconomic trends towards sustainable development (SSP1), in which there are low challenges to mitigation and adaptation, avoid 0.53 million (ranging from 0.49 – 0.58) deaths which is nearly double the 0.26 million (0.14 – 0.39) deaths avoided in fragmentation (SSP3) in which there are high challenges to mitigation and adaptation. Conventional development (SSP5) and middle-of-the-road development (SSP2) have health benefits between these two extremes: 0.47 (0.38 – 0.55) and 0.41 (0.35 – 0.53) million, respectively, whereas trends towards inequality (SSP4) more closely match fragmentation with 0.28 (0.19 – 0.40) million deaths avoided. Overall, while low challenges to both mitigation and adaptation have health benefits, our estimates suggest that challenges to adaptation are especially connected to the deaths avoided from climate action.

[110-119]

In the most optimistic scenario (SSP1-19) in which there is sustainable socioeconomic development and a strong reduction in climate forcing that presents low challenges to mitigation and adaptation, emission reductions could avoid 1.32 million deaths (with a lower and upper bound of 0.95, 1.73). This is over four times the 0.32 million (0.24, 0.42) deaths avoided in SSP3-60, in which there is fragmented socioeconomic development, weak reductions to climate

forcing, and high challenges to mitigation and adaptation. Conventional development (SSP5) and middle-of-the-road development (SSP2) towards RCP-45 result in climate co-benefits between these two extremes of 1.03 (0.75, 1.35) and 0.79 (0.58, 1.04) million deaths avoided, respectively, whereas trends towards inequality (SSP4) for RCP-45 closely match fragmentation with 0.51 (0.37, 0.67) million deaths avoided.

We also note that due to an error in our original analysis the magnitudes of the co-benefits were underestimated hence the higher values in the updated text (see section 1.0 for details).

R2C7: *The concept of the imbalance ratio is introduced around line 131, but its calculation is not explained until the Methods section (line 531 onward), and the visual schematic appears only in the supplement (Figure S1a).*

Reviewer #1 had a similar comment (R1C3); we have addressed this by including the schematic in the main text as the new Figure 1 so that it is presented before we begin interpreting the indicators. Please refer to that comment for detailed changes related to this issue.

R2C8: *In Figure S5, the “Asia to Europe” exchange ratio under one of the SSP-RCP scenarios (likely around SSP2-34 or SSP2-45) spikes to a value exceeding 5, which is visually inconsistent with the rest of the dataset and other scenario curves. This point is not acknowledged or explained in the figure caption, nor is there any indication that the y-axis was clipped or rescaled to accommodate outliers.*

Thank you for pointing out that we neglected to comment on this in the original text. We have updated the supplemental figures in the styles of the main text figures now, so this visual inconsistency is no longer plotted. Out of curiosity we looked into this, and this is attributable to unique conditions in SSP2-45 in that major emission changes occur for India in 2030 (relative to SSP3-Baseline) but not as much for other countries.

R2C9: *The caption for Figure 4 explains that values greater than 1 mean Region A contributes more to Region B’s benefits than vice versa. However, this phrasing could be misread as “Region A receives more benefit” rather than “Region A’s actions have greater effect”. Rephrase the caption to clarify directionality, e.g., “Values greater than one indicate that climate mitigation in Region A provides more benefits to Region B than the reverse.”*

We have now updated this figure and made major changes to this caption.

R2C10: *Table 1 presents NO_x emissions in teragrams and avoided deaths in thousands, but these units are not explicitly listed in the table headers. I think add units in parentheses to the headers for clarity (e.g., “Global NO_x Emissions (Tg)”, “Global Deaths Avoided (Thousands)”) will improve current version.*

Thank you for this suggestion; we have made this change (see section 4.0).

R2C11: *Oceania is excluded from Figure 1c on the grounds of having disproportionately high imbalance ratios due to small population size. However, no supporting data is shown to validate*

this exclusion. Please provide numerical values or variance ranges for Oceania in the supplement to justify its removal from the main analysis.

Thank you for this point. In the new metrics we no longer have this issue as we include Oceania data in Figure 1 now; however, out of curiosity we plotted the old imbalance ratios and include them below for reference:

3.3 Reviewer #3:

R3C0: *I co-reviewed this manuscript with one of the reviewers who provided the listed reports. This is part of the Nature Communications initiative to facilitate training in peer review and to provide appropriate recognition for Early Career Researchers who co-review manuscripts.*

Thank you for your contributions to the revision of this manuscript. We appreciate the time that you took to review and develop comments on our paper.

3.4 Reviewer #4:

R4C0: *This study involves a substantial amount of work. It presents regional inequality in the benefits of emission reduction under different SSP and RCP scenarios from a novel perspective, while also considering transboundary pollution transport. However, several issues still require the authors' attention. Please make major revisions to the article.*

Thank you for your positive feedback and useful comments; we have endeavored to incorporate your suggestions and believe that they have improved the quality of the manuscript.

R4C1: *The abstract is difficult to understand without reading the full manuscript. It requires appropriate revision for clarity.*

We agree with you that the original abstract was difficult to interpret without reading the full manuscript (partially due to the confusing old indicators). We have revised it to improve interpretability:

(18-32)

Climate action improves air quality and health by reducing sources that co-emit greenhouse gases and air pollutants such as PM_{2.5}, yet ill-conceived mitigation could induce imbalances in the distribution and exchange of air pollution across international borders. Despite its potential to endanger equality, the future effects of climate action on transboundary PM_{2.5} are largely unstudied. Here, we show that health co-benefits are maximized through sustainable development (581 thousand deaths avoided); however, imbalances in the transboundary exchange of PM_{2.5}-related health impacts vary across different climate-related emission scenarios and regions. In sustainable futures, African countries contribute more co-benefits to Europe than vice-versa (805 net deaths avoided); however, in more fragmented futures this dynamic reverses (-356 net deaths avoided) and African countries are more dependent on European climate action. This variation in transboundary air pollution across different climate futures underscores the need to characterize how climate action – or lack thereof – could ameliorate, perpetuate, or exacerbate air pollution inequality beyond solely quantifying health co-benefits. Future climate-related policy design could consider these effects to rectify current and historical environmental injustices.

[18-33]

Climate action redresses poor air quality and its health burden by reducing sources that co-emit greenhouse gases and air pollutants such as PM_{2.5}, yet ill-conceived mitigation could induce imbalances in the distribution and exchange of air pollution across international borders. Despite its potential to endanger equality, the future effects of climate action on transboundary PM_{2.5} are largely unstudied. Air quality co-benefits are maximized through sustainable development towards RCP-19 (1.32 million deaths avoided) and minimized in fragmentation trends towards RCP-60 (0.32 million). Futures with stronger climate mitigation necessitate more aggressive climate action from developed nations that increases the fraction of co-benefits that are transboundary for Africa, ranging from 8% higher in SSP1 to 53% in SSP3. Additionally, countries with lower human development indices have a greater fraction of their co-benefits originate externally (0.76 in SSP1-26) than more developed countries (0.65) which positions developing nations as more dependent on regional and global climate action than developed countries. Co-benefit exchanges between countries are non-uniform, which suggests country-specific analyses are necessary to develop climate policy that minimize transboundary effects. These results support the need for nuanced policy design that considers how transboundary co-benefits respond to regional differences, distinct socioeconomic trends, and mitigation strategies.

R4C2: *Table 1 does not present all scenarios, such as SSP3-Ba. The authors should clarify the reason for this omission.*

Thank you for bringing this up, we include all scenarios in Table 1 except the SSP3-Baseline which we use as a reference point to calculate our health impacts. However, this was not explained originally and so we have updated the Table 1 caption to reflect this:

(111-112)

Table 1. Global total NO_x emissions in 2040 (Tg), global deaths avoided in 2040, narrative descriptions, and challenges for all scenarios considered in this study.

[139-145]

Table 1. Details on the 24 future climate emission scenarios considered in this study across five shared socioeconomic pathways (SSPs) and six representative concentration Pathways (RCPs). Global NO_x emissions are presented in teragrams along with the global deaths avoided from reductions in PM_{2.5}, with lower and upper bound health estimates included in brackets. The narratives and challenges associated with the SSPs are also included. All scenarios used in our analysis are presented here with the exception of SSP3-Baseline, which is used as the baseline against which we calculate the climate co-benefits.

R4C3: *The reference to Figure 1 in line 116 seems inappropriate. In addition to providing Table S1, a spatial distribution map of the six regions should also be included.*

We agree that the reference to Figure 1 (now Figure 2) was inappropriate in its original location and have now updated this:

(115-117)

To further explore how socioeconomic trends and mitigation strategies vary across different geographies, we group all countries into one of six regions: South America (SA), Oceania (OC), North America (NA), Europe (EU), Asia (AS), and Africa (AF) (Figure 1).

[178-180]

For the six regions we consider in this study, scenarios in which climate action is more aggressive (i.e., lower RCPs and SSPs with fewer challenges) achieve greater climate co-benefits (Figure 2a).

Additionally, we have taken your suggestion and included a spatial distribution map of the six regions in the supplement (Figure S1) that we have also included in section 4.0 of this document.

R4C4: *Why are Europe and Africa grouped into the same category? Is this classification unrelated to socioeconomic development levels? The rationale should be further explained.*

Apologies, we are not entirely sure which aspect of our original analysis this refers to; however, we believe that you are referring to what was originally presented in Figure 4 in which we calculated exchange ratios between Europe and Africa. We have now removed this figure from our analysis and so we believe this is no longer an issue.

R4C5: *What does “IR” refer to in line 223? Its full name should be provided upon first appearance.*

This “IR” originally referred to the “imbalance ratio” but we no longer use this term in this work to simplify our analysis (see R1C3 for details) and so this has been removed.

R4C6: *What is the reason for the inclusion of the four specific regions in Figure 2a? Please clarify.*

Originally, Figure 2a referred to the scenario in which co-benefits were maximized (generally SSP1) but we have now overhauled this figure in response to your next comment and so we believe that this issue has now been resolved.

R4C7: *The results presented in the manuscript are relatively limited. It is recommended to include spatial distribution and flow maps of health co-benefits under different scenarios.*

Thank you for this suggestion, we have endeavored to address this by updating all of our figures to provide more detail related to the inequalities borne by transboundary air pollution across the different climate futures. Specifically, we have dramatically changed Figures 3 and 5 (formerly Figures 2 and 4) and specifically with the prior have updated it to better capture the spatial distribution related to co-benefits, as you suggest. We have included all of the new main text figures in section 4.0 of this document. We also experimented with developing flow maps of health co-benefits; however, given the amount of country relationships we consider, this looked messy, so we decided not to include them in this version of the manuscript and instead use heatmaps to depict co-benefit exchanges in Figure 4.

R4C8: *The manuscript primarily provides descriptive explanations of the observed phenomena, but lacks further analysis of the underlying causes. What specific factors lead to regional inequality in climate action benefits under different scenarios? Are these related to economic levels or population structures? Moreover, are there any suggestions for more equitable mitigation scenarios or compensatory mechanisms?*

Thank you for bringing up this important point. These echoes sentiments brought up by the other reviewers (for suggestions on more equitable scenarios or policies please refer to R2C2) and we have sought to update our manuscript to provide fewer descriptive explanations and more quantitative analysis and have used these to discuss possible policy applications. We have also now discussed the underlying causes of the observed phenomena and the specific factors leading to regional inequality and specifically comment on economic levels (including GDP in Figure 5) and population:

(107-109)

Overall, while low challenges to both mitigation and adaptation have health benefits, our estimates suggest that challenges to adaptation are especially connected to the deaths avoided from climate action.

[121-137]

There are two compounding factors that drive variability across these different climate futures: the degree of mitigation and type of socioeconomic development. Stronger mitigation (i.e., lower

RCPs) engenders greater co-benefits; however, the strength of this effect is mediated by the socioeconomic development pathway in which the action is implemented. Specifically, the effect of stronger mitigation is relatively weak in sustainable development – there are 1.1 times as many co-benefits in SSP1-26 compared to SSP1-45 – and stronger in more fragmented climate futures – there are 2.0 times as many co-benefits in SSP3-26 compared to SSP3-45. Similarly, socioeconomic development that is more equitable and fosters global cooperation generally leads to greater co-benefits compared to less equitable socioeconomic development, but this is dependent on the degree of climate mitigation. For example, for RCP-26, SSP1 results in 1.4 times as many co-benefits as SSP3 while in RCP-45 SSP1 results in 2.4 times as many co-benefits as SSP3. These results suggest that climate policy would ideally be designed to consider socioeconomic development alongside the degree of mitigation given these dependencies. Ultimately, in climate futures for which there are low challenges to both mitigation and adaptation the improvement to air quality will have substantial health benefits; however, our estimates suggest that challenges to adaptation are especially connected to the deaths avoided from climate action.

(127-129)

However, in fragmented futures, such as SSP3-60, the Asian share of co-benefits grows (83%) and the share of other regions decreases relatively for Europe (8%), Africa (5%), North America (3%), and South America (1%).

[184-192]

In SSP3-60 the Asian share of co-benefits grows (85%) while other regions such as Europe (7%), Africa (3%), North America (4%), and South America (1%) benefit proportionally less. This high share of co-benefits in Asia is not surprising and is attributable to two factors: (1) the population of Asia is large at present and by 2040 this young population will age and become more susceptible to PM_{2.5}-related health impacts and (2) China and India are developing rapidly and these two countries receive a majority of the PM_{2.5}-related deaths in the region, thus climate action in these countries could alleviate a great degree of the air pollution health burden. Specifically, co-benefits in these two countries alone make up between 52% (SSP1-19) and 63% (SSP3-60) of all global co-benefits.

[266-280]

For example, in SSP3-60 removing these two countries leads to a higher global transboundary fraction of 0.54. This explains why when calculating the average transboundary fraction across all countries there is a substantially higher value of 0.68 (SSP3-60) than the total global transboundary fraction. We note that these transboundary contributions are substantially greater than past analyses^{16,17} of historical transboundary air pollution burdens, which found 14% and 12% of all PM_{2.5}-related deaths were attributable to transboundary pollution. This difference is likely attributable to three main factors: (1) these past studies focused on aggregated groups of countries whose boundaries are further from peak emissions locations whereas we perform this calculation for individual countries, (2) in this study we consider climate action whereas these previous studies consider all anthropogenic emissions; climate action is not adopted to the same degree globally and many of the highest emission reductions are implemented in a small set of developed nations, and (3) many of the targets of climate action are precursors of secondary

PM_{2.5} (specifically NO_x and SO₂) and not primary PM_{2.5} and thus this action targets pollution that is more likely to be transported greater distances due to its longer atmospheric lifetime.

[401-409]

Contrasting this with the transition from a middle-of-the-road to sustainable climate future (Figure 5c), the exchange between Africa and Europe is essentially unaffected (from 29% to 28%, African favored). This suggests that fragmentation contributes to a balanced Africa and Europe exchange and through trends towards sustainable or middle-of-the-road development, Africa contributes more to Europe than vice versa due to a greater capacity to adopt more aggressive climate action. Generally, at the regional scale, changes to exchanges were mostly unaffected by this socioeconomic development excluding the exchange between South America and Africa which becomes less African dominated in sustainable development compared to both SSP3-45 and SSP2-45.

[517-528]

The transition to more sustainable development does not always reduce inequality, e.g., for France and Spain the exchange in SSP3-26 is more balanced (3.67) than compared to SSP1-26 (4.13). Ultimately, there is not a uniform relationship between socioeconomic development and exchange inequalities: the same transition could ameliorate some exchange inequalities while perpetuating or exacerbating others. However, generally a transition from a more fragmented climate future to a more sustainable ones lead to greater balance: comparing SSP1-45 to SSP3-45, nine of the ten largest shifts in exchanges between countries enabled more balanced exchanges (i.e., where pairs of countries contribute even co-benefits to one another) and many of these shifts were towards greater contributions from the lower GDP. This occurs because sustainable development fosters greater global cooperation; developing nations develop and industrialize quicker and thus they have more potential to control emissions.

[392-409]

Comparing fragmented socioeconomic development to sustainable socioeconomic development results in some of the largest changes to exchange inequalities between regional and country pairs. For example, in SSP3-45, the exchange between Europe and Africa is relatively balanced: the prior contributes 48% of the total co-benefits exchanged and the latter, 52% (Figure 5a). Transitioning to more sustainable socioeconomic development leads to Africa – the region with the weaker GDP today – contributing a clear majority of co-benefits to the exchange (72%). This shift towards greater African contributions to Europe in a more sustainable world occurs because a more sustainable climate future necessitates growth in historically underdeveloped regions and enables them to contribute more to climate action and subsequently greater contributions to co-benefits. Contrasting this with the transition from a middle-of-the-road to sustainable climate future (Figure 5c), the exchange between Africa and Europe is essentially unaffected (from 29% to 28%, African favored). This suggests that fragmentation contributes to a balanced Africa and Europe exchange and through trends towards sustainable or middle-of-the-road development, Africa contributes more to Europe than vice versa due to a greater capacity to adopt more aggressive climate action. Generally, at the regional scale, changes to exchanges were mostly unaffected by this socioeconomic development excluding the exchange between South America and Africa which becomes less African dominated in sustainable development compared to both SSP3-45 and SSP2-45.

4.0 Tables and Figures

Table 1. Details on the 24 future climate emission scenarios considered in this study across five shared socioeconomic pathways (SSPs) and six representative concentration Pathways (RCPs). Global NO_x emissions are presented in teragrams along with the global deaths avoided from reductions in PM_{2.5}, with lower and upper bound health estimates included in brackets. The narratives and challenges associated with the SSPs are also included. All scenarios used in our analysis are presented here with the exception of SSP3-Baseline, which is used as the baseline against which we calculate the climate co-benefits.

SSP	RCP	Global NO _x Emissions (Tg)	Global Deaths Avoided (Millions) [Uncertainty]	Narrative	Challenges
1	19	24.0	1.32 [0.95, 1.73]	Sustainability	Low challenges to adaptation and mitigation
	26	31.8	1.26 [0.90, 1.64]		
	34	43.7	1.16 [0.83, 1.52]		
	45	52.3	1.11 [0.80, 1.45]		
	Ba	55.4	1.08 [0.78, 1.42]		
2	19	33.7	1.22 [0.88, 1.59]	Middle of the Road	Intermediate challenges to adaptation and mitigation
	26	45.8	1.13 [0.81, 1.47]		
	34	62.8	0.98 [0.71, 1.28]		
	45	85.1	0.79 [0.58, 1.04]		
	60	90.5	0.75 [0.55, 0.99]		
	Ba	102.4	0.67 [0.49, 0.87]		
3	26	61.9	0.90 [0.66, 1.17]	Fragmentation	High challenges to adaptation and mitigation
	34	80.5	0.69 [0.51, 0.90]		
	45	102.1	0.46 [0.34, 0.60]		
	60	113.0	0.32 [0.24, 0.42]		
4	26	57.8	0.91 [0.66, 1.18]	Inequality	High challenges to adaptation low challenges to mitigation
	34	79.0	0.69 [0.51, 0.91]		
	45	98.1	0.51 [0.37, 0.67]		
	Ba	107.1	0.42 [0.30, 0.55]		
5	26	33.2	1.26 [0.91, 1.64]	Conventional Development	Low challenges to adaptation high challenges to mitigation
	34	47.0	1.14 [0.82, 1.48]		
	45	58.3	1.03 [0.75, 1.35]		
	60	62.0	1.00 [0.72, 1.30]		
	Ba	83.9	0.82 [0.60, 1.07]		

a**b**
Figure 1. Schematics of the metrics used to quantify inequalities associated with transboundary air pollution across different climate futures. (a) Transboundary fractions (TF) characterize the extent of co-benefits in a country that come from external action; co-benefits are the premature deaths avoided owing to reduced PM_{2.5} concentrations. (b) Exchanges (EXC) compare co-benefits from emission reductions between a pair of countries or regions; the contribution to total benefits exchanged (TEC) indicates how much a country contributes to the total gross co-benefits exchanged between two countries.

Figure 2. (a) The total deaths avoided for each of the SSP-RCP scenarios relative to SSP3-Baseline in 2040 ordered from the most to least equitable scenarios. Colors indicate the receptor region in which the co-benefits occurred (i.e., where the deaths were avoided). Socioeconomic trends and mitigation strategies are included on the y-axis. Error bars refer to the lower and upper bound uncertainty from the health impact assessment for the total co-benefits. (b) The co-benefits specifically attributable to external action for each of the receptor regions (i.e., the transboundary co-benefits). The fraction of co-benefits that are transboundary broken down by (c) socioeconomic development type and (d) mitigation strategy.

Figure 3. Transboundary fractions for individual countries in a sustainable and strong climate mitigation scenario (SSP1-26) (top left) and the percent difference in transboundary fractions for each country for a weaker RCP (SSP1-45) (top right), a less equitable SSP (SSP3-26) (bottom left) and a scenario that has both weaker climate mitigation and less equal socioeconomic development (SSP3-45) (bottom right).

Figure 4. Exchanges (EXC) of climate co-benefits within and between Europe, Africa, and Asia for SSP1-26 in 2040 (left). Darker colors indicate greater co-benefits in the receptor country attributable to emission reductions in the source country; self-contributions (i.e., the diagonal) are excluded. Asian exchanges are a factor of ten greater than exchanges in Africa and Europe and the color-scales are logarithmic. Heatmaps of transboundary exchanges of climate action within Europe, Africa, and Asia relative to SSP1 in 2040 for RCP-26 for SSP2, SSP3, SSP4, and SSP5 (right). The color-scales are linear. Interregional exchanges (i.e., Africa to Europe and Asia to Africa) are provided in an absolute sense – not relative to SSP1.

Figure 5. The change in the percentage of co-benefits exchanged (TEC × 100%) between (a) regional and (b) country pairs that is contributed by the higher GDP (first name) from SSP3-45 to SSP1-45 in 2040. For the regional exchanges, the arrow points from the less equal scenario (SSP3-45) to the more equal scenario (SSP1-45) and the color of the dot indicates the higher regional contributor. For the country exchanges, the arrow is the same but the black dot represents SSP3-45 and the white dot represents SSP1-45 and the background shading indicates in which region the exchange occurs as labelled above Figure 5a; grey indicates exchange between different regions. Panels (c) and (d) are the same as (a) and (b), respectively, but for middle-of-the-road development (SSP2 vs SSP1) instead of fragmentation development.

Figure S1. Regional assignments for all countries considered, as either a source or receptor, in this work in map form.

Figure S2. Transboundary fractions for individual countries for a sustainable and strong climate forcing scenario (SSP1-26) in 2030 (top left) and the percent difference in transboundary fractions for each country for the same scenario in 2040 (SSP1-26) (top right), a less equitable SSP in 2030 (SSP3-26) (bottom left) and the less equitable SSP in 2040 (bottom right).

5.0 References

1. Nawaz, M. O. & Henze, D. K. Premature Deaths in Brazil Associated With Long-Term Exposure to PM_{2.5} From Amazon Fires Between 2016 and 2019. *GeoHealth* **4**, e2020GH000268 (2020).
2. Fujimori, S., Hasegawa, T., Ito, A., Takahashi, K. & Masui, T. Gridded emissions and land-use data for 2005–2100 under diverse socioeconomic and climate mitigation scenarios. *Sci. Data* **5**, 180210 (2018).
3. Hoesly, R. M. *et al.* Historical (1750–2014) anthropogenic emissions of reactive gases and aerosols from the Community Emissions Data System (CEDS). *Geosci. Model Dev.* **11**, 369–408 (2018).
4. Heald, C. L. *et al.* Atmospheric ammonia and particulate inorganic nitrogen over the United States. *Atmospheric Chem. Phys.* **12**, 10295–10312 (2012).
5. Henze, D. K., Seinfeld, J. H. & Shindell, D. T. Inverse modeling and mapping US air quality influences of inorganic PM_{2.5} precursor emissions using the adjoint of GEOS-Chem. *Atmospheric Chem. Phys.* **9**, 5877–5903 (2009).
6. Philip, S. *et al.* Anthropogenic fugitive, combustion and industrial dust is a significant, underrepresented fine particulate matter source in global atmospheric models. *Environ. Res. Lett.* **12**, 044018 (2017).
7. Nault, B. A. *et al.* Secondary organic aerosols from anthropogenic volatile organic compounds contribute substantially to air pollution mortality. *Atmospheric Chem. Phys.* **21**, 11201–11224 (2021).
8. Thunis, P. *et al.* Non-linear response of PM_{2.5} to changes in NO_x and NH₃ emissions in the Po basin (Italy): consequences for air quality plans. *Atmospheric Chem. Phys.* **21**, 9309–9327 (2021).
9. Wiel, K. van der, Selten, F. M., Bintanja, R., Blackport, R. & Screen, J. A. Ensemble climate-impact modelling: extreme impacts from moderate meteorological conditions. *Environ. Res. Lett.* **15**, 034050 (2020).
10. Clarke, B., Otto, F., Stuart-Smith, R. & Harrington, L. Extreme weather impacts of climate change: an attribution perspective. *Environ. Res. Clim.* **1**, 012001 (2022).
11. Turnock, S. T. *et al.* Historical and future changes in air pollutants from CMIP6 models. *Atmospheric Chem. Phys.* **20**, 14547–14579 (2020).
12. Kumar, P. Climate Change and Cities: Challenges Ahead. *Front. Sustain. Cities* **3**, (2021).
13. Nault, B. A. *et al.* Anthropogenic Secondary Organic Aerosols Contribute Substantially to Air Pollution Mortality. <https://acp.copernicus.org/preprints/acp-2020-914/> (2020) doi:10.5194/acp-2020-914.
14. Nansai, K. *et al.* Consumption in the G20 nations causes particulate air pollution resulting in two million premature deaths annually. *Nat. Commun.* **12**, 6286 (2021).
15. Fang, D. & Chen, B. Inequality of air pollution and carbon emission embodied in inter-regional transport. *Energy Procedia* **158**, 3833–3839 (2019).
16. Chen, L. *et al.* Inequality in historical transboundary anthropogenic PM_{2.5} health impacts. *Sci. Bull.* **67**, 437–444 (2022).
17. Zhang, Q. *et al.* Transboundary health impacts of transported global air pollution and international trade. *Nature* **543**, 705–709 (2017).

18. Renna, S., Granella, F., Aleluia Reis, L. & Schulz-Antipa, P. CLAQC v1.0 – Country Level Air Quality Calculator: an empirical modeling approach. *Geosci. Model Dev.* **18**, 2373–2408 (2025).
19. Eastham, S. D., Monier, E., Rothenberg, D., Paltsev, S. & Selin, N. E. Rapid Estimation of Climate–Air Quality Interactions in Integrated Assessment Using a Response Surface Model. *ACS Environ. Au* **3**, 153–163 (2023).
20. Zhang, Q. *et al.* Transboundary health impacts of transported global air pollution and international trade. *Nature* **543**, 705–709 (2017).
21. Nawaz, M. O. *et al.* A Source Apportionment and Emission Scenario Assessment of PM2.5- and O3-Related Health Impacts in G20 Countries. *GeoHealth* **7**, e2022GH000713 (2023).
22. Nawaz, M. O. *et al.* A Source Apportionment and Emission Scenario Assessment of PM2.5- and O3-Related Health Impacts in G20 Countries. *GeoHealth* **7**, e2022GH000713 (2023).
23. United Nations Development Programme. Human Development Index (HDI). (2025).

Response to Reviewers

1.0 Guidance for this document:

We thank the editorial team and the reviewers for their second review of our manuscript entitled “*National climate action can ameliorate, perpetuate, or exacerbate international air pollution inequalities*” that we submitted to *Nature Communications* earlier this year. We have endeavored to integrate this second round of comments. To differentiate text from the reviewers, the authors, and new additions to the text please refer to the key:

Element	Text Style
Reviewer Comment	Italicized black text
Author Comment	Plain black text
Original Manuscript Text	Plain green text
Revised Manuscript Text	Plain blue text
Original Line Number	(in parentheses)
New Line Number	[in brackets]

All of the figures that are mentioned in response to reviewer comments can be found at the end of this document.

2.0 Reviewer Comments

2.1 Reviewer #1:

RIC0: *I appreciate the authors' detailed response, and most of the issues have been adequately addressed.*

Thank you for spending the additional time that you have taken to review our revised manuscript and for your positive comment.

RIC1: *Regarding the uncertainty in the results, the authors acknowledge in their discussion that uncertainties arise from various aspects such as emission projections, chemical transport model simulations, and adjoint sensitivity calculations. However, the current revision still limits the quantification of uncertainty mainly to the health impact calculations. In the GEOS-Chem analysis, simply stating that "transboundary pollution may be underestimated" is insufficient. Readers may want to know what this implies for the final estimates of mortality. This work does not present major innovations about the methodological framework. Extensive literature has already discussed the uncertainty of chemical transport models and the propagation of model uncertainties. It is suggested that the authors refer to some methodological studies to provide a more substantive quantification of the uncertainties in their results.*

Thank you for raising this issue and we agree with you that in the revised manuscript most of the quantitative discussion of uncertainty is on the health impact calculation. Given that similar methodological frameworks have been used before, as you suggest, we can better quantify uncertainty in the other steps of our analysis by referencing past studies. To address this, we have revised the uncertainty analysis section, replacing qualitative descriptions of uncertainty in our analysis with quantitative ones. Specifically, we have added additional detail about how our emissions compare to other bottom-up inventories, included uncertainty estimates of the GEOS-Chem adjoint of PM_{2.5} and its related health burden from HTAP analyses, identified the uncertainty introduced by the linear adjoint calculation from a past study projecting PM_{2.5} from SSP emission scenarios, and estimated how the absence of SOA could bias our results and transboundary fractions. In response to R4C8 we have also rephrased some of the text to be less repetitive and more concise. We have included the updated uncertainty analysis section below:

[890-960]

Uncertainty is introduced primarily by: (1) emission projections, (2) the GEOS-Chem forward simulation, (3) the adjoint sensitivity calculation, and (4) the health impact analysis.

Although it is impossible to quantify a specific uncertainty associated with emissions projected for the future – given that the true values are unknown – the emission projections⁴⁹ were tested against historical emissions from another inventory, the Community Emissions Data System (CEDS)⁵⁰ and they were found to be well correlated for SO₂ from all sectors in 2005 ($R^2=0.71$). Additionally, the correlation with CEDS projections in 2050 for the energy sector was examined for both SO₂ ($R^2=0.44$) and NO_x ($R^2=0.70$). The HTAP emissions⁵¹ that drive our forward simulation have been compared to other inventories⁵². Of the twelve regions considered in that analysis, HTAP had the minimum emissions of all bottom-up inventories in five of the regions for NO_x (27% lower than average), none of the regions for SO₂, five of the regions for BC (44%

lower than average), and three of the regions for OC (48% lower than average). Additionally, HTAP had the maximum emissions in none of the regions for NO_x, two of the regions for SO₂ (39% higher than average), two of the regions for BC (21% higher than average), and two of the regions for OC (62% higher than average). This comparison suggests that the HTAP emissions have regional biases; however, compared to CEDS, the HTAP emissions of NO_x are approximately 16% lower on a global scale. Given the degree of uncertainty in these emission projections, we exclusively consider the relative differences in projections as opposed to absolute differences for our analysis.

GEOS-Chem has a strong record of estimating atmospheric composition that compares well to observations^{53,54}. In an HTAP ensemble analysis⁵⁵ of the PM_{2.5} health burden associated with intercontinental transport, the simulated PM_{2.5} of the GEOS-Chem adjoint was compared to over 3000 global monitors. GEOS-Chem adjoint simulated concentrations fell within the range of ensemble members with a normalized mean error of 55.1% (ranged from NME=35.4% to NME=62.9%) and a correlation of 0.65 (ranged from R=0.63 to R=0.77). The GEOS-Chem adjoint had the most positive bias of the ensemble members (NMB=20.3%); however, in an absolute sense the bias was in the middle of the spread of ensemble members (ranged from NMB=-60.9% to NMB=20.3%). Additionally, all of the ensemble members underestimated the health burden of PM_{2.5} compared to the GBD 2015 study (4.2 million premature deaths); thus, this potential high bias in the GEOS-Chem adjoint led to a more accurate estimate of health impacts (3.2 million premature deaths) than the multi-model mean (2.8 million premature deaths). We also note that a majority of the surface-level observations were located in North America, Europe, and China; thus, this analysis may not be representative of biases in other regions of the world.

The version of GEOS-Chem that is used in our analysis does not include anthropogenic fugitive dust and secondary organic aerosol; the prior was estimated⁵⁶ to increase global population-weighted PM_{2.5} by around 2.9 μg m⁻³ and the latter was estimated³⁶ to make up between 15%-30% of anthropogenic PM_{2.5} in urban environments. Fugitive dust PM_{2.5} is a primary species with a short atmospheric lifetime that mainly affects concentrations near where it is emitted. In contrast, secondary organic aerosol has a longer atmospheric lifetime enabling its transport across greater distances. For example, one study³⁸ simulated PM_{2.5} transport for one UK city, and found that approximately 5% of the total PM_{2.5} came from non-local secondary organic aerosol. In our previous work³⁷, we estimated that around 10% of PM_{2.5} in DC, originated from non-local secondary organic aerosol. These past analyses suggest that the exclusion of SOA could lead to an underestimate of between 5 – 10% of non-local PM_{2.5} that in turn would imply an underestimate of transboundary factions and exchanges by this magnitude. However, we note that these past studies are for two locations in the Global North and not representative of the regional variation in this study and that the SOA contributions are calculated for urban areas that have higher local sources than rural areas.

Another source of uncertainty in our analysis is the application of local-linear sensitivities to inherently non-linear PM_{2.5} formation. Although the primary components of PM_{2.5} (i.e., black and organic carbonaceous aerosol) respond linearly to emissions, secondary inorganic aerosol does not respond linearly. Thus, this approach does not capture the second order effects of secondary inorganic aerosol formation. A previous study³¹ explored the uncertainty associated

with this local-linear assumption in response to SSP emission projections in Korea, and found that non-linearities contributed an underestimate of up to 57% in the PM_{2.5}-related health benefits from mitigation (although this uncertainty was lower for 2040).

Additionally, we have made some minor changes to the discussion to reflect the new quantitative numbers from the uncertainty analysis

[650-668]

The results presented in our work should be considered alongside sources of uncertainty. First, our adjoint-derived methodology calculates linear relationships between emissions and PM_{2.5} and while some components of PM_{2.5} (e.g., primary carbonaceous aerosol) are formed linearly, others (e.g., secondary inorganic aerosol) are formed through complex non-linear chemistry³⁰; previous work³¹ has suggested that this non-linearity could lead to underestimates of PM_{2.5}-related health impacts up to 57%. Additionally, our estimates of co-benefit contributions rely on the simplified assumption that the pollution formation relationships that we calculate for our base year (i.e., 2010) will be maintained in the future. We anticipate that changes to climate^{32,33} and chemical^{34,35} environment in the future will modify transboundary air pollution and we suggest that feedbacks between climate change and transboundary air pollution are explored in future work. Our analysis neglects to include anthropogenic fugitive dust and secondary organic aerosol formation that contribute to PM_{2.5}, especially in heavily populated urban areas³⁶. Previous studies^{37,38} indicate that for urban locations, between 5 and 10% of PM_{2.5} originates from non-local SOA. We suggest that future work is done to quantify the transboundary fractions of SOA globally.

R1C2: *And, there is a minor issue that the equation on line 725 appears to be incomplete.*

Thank you for catching this mistake. This has now been fixed:

(725)

Before:

$$EXC_{m,n,s} = \tag{6}$$

[870]

After:

$$EXC_{m,n,s} = \sum_{I \in m} \Delta J_{I,n,s} \tag{6}$$

2.2 Reviewer #2:

R2C0: *The figures play a central role in communicating the results; however, I have identified several issues regarding figure formatting, consistency, and clarity that require substantial revision prior to publication. Addressing these issues will substantially improve the quality and impact of the visual presentation in this manuscript. Below, I provide detailed comments for each figure:*

Thank you for your detailed comments on our manuscript. We agree that through improving the formatting, consistency, and clarity of our figures we can increase the quality of our manuscript, and we appreciate your suggestions on this matter. Please see our responses below:

General Figure Issues:

- 1. There is a lack of consistency in font style and size across figures, with some using Arial and others Times New Roman.*
- 2. Legends and labels are sometimes incomplete or formatted in a non-standard way.*
- 3. Axis labels, panel annotations, and units are not always clearly presented.*
- 4. In-figure legends and explanations are missing in some multi-panel figures, making interpretation less accessible.*

We appreciate your summary of the main issues with the figures. In response to your comments, we have now updated the figures to use a consistent font type (i.e., Arial). We have adjusted the legends and labels based on your suggestions (please refer to our figure-specific comments below for details). Additionally, we have included the axis labels, annotations, and units based on your suggestions and included in-figure legends to improve interpretability. For specific changes, please see our responses below:

R2C1: *Figure 2: 1. All textual elements, including axis labels, titles, legends, and color bars, are set in Times New Roman, while Figures 1, 4, and 5 use Arial. I recommend standardizing all figures to use Arial and adopting consistent font sizes. For example, use 14 pt for titles and 12 pt for axis labels and legends throughout the manuscript; 2. The color legend for continents in panels (a) and (b) does not follow standard formatting. Please revise using standard legend boxes with clear labels for each region; 3. The x-axis in panel (c) is incomplete, as only partial scenario labels such as 19 and 26 are shown. This may be confusing to readers. Please ensure that all axis labels and tick marks are fully represented.*

Thank you for your detailed comments on Figure 2. First, we have taken your suggestion here and changed the figure font throughout from Times New Roman to Arial; in doing so, we now have a consistent font type in our manuscript figures. In terms of the font size, we have now standardized this to use 14 pt for titles, 12 pt for axis labels and legends, and 10 pt for tick labels. We have also changed the color legend to follow standard formatting and put this in a box in Figure 2b. For the x-tick labels for panel (c), we now include “RCP” in front of the RCP number to avoid confusing readers. We appreciate your last comment; however, we have opted to group similar panels together to avoid unnecessary text on the figures and to enable quick comparison between different scenario. Specifically, given that the panels share a common y-axis, we hide

the y-labels and y-tick marks to give the figure a cleaner appearance. For your reference, we have included the updated version of Figure 2 at the bottom of this document.

R2C2: *Figure 3: 1. All elements in this figure are presented in Times New Roman. For consistency, please change all fonts to Arial so that all figures follow the same style; 2. The upper color bar does not include a unit. Please add “Transboundary Fraction (unitless)” or “Transboundary Fraction (%)” as appropriate. For the lower color bars, make sure the unit “percentage (%)” is clearly indicated; 3. Currently, the figure does not display panel labels or scenario flow arrows. I recommend adding clear panel labels, such as (a), (b), (c), and (d), in the upper left of each map. It would also be helpful to include arrows to illustrate the direction of scenario changes; 4. The figure caption should be expanded to provide a distinct explanation for each panel, including a description of the scenarios being compared and the direction of change.*

In response to R2C1, we have now updated the font types of all figures from Times New Roman to Arial to maintain consistency. Thank you for pointing out that the units of the colorbars are unclear; we have added “unitless” to the colorbar label and rewritten the other labels to more clearly indicate the “%”. We agree that adding panel labels enables us to describe the specific scenarios more clearly in the figure caption; so, based on your suggestion, we have now included panel labels. We appreciate the suggestion to include arrows to illustrate the direction of scenario changes; however, we have opted to demonstrate the direction of the scenario changes on the subplots directly (e.g., “SSP3-45 – SSP1-26”) to indicate the difference between the two scenarios. We have now also mentioned this explicitly in the caption and, based on your earlier comment, we have updated the panel figure captions to describe this for each subplot. For your reference, we have included the updated version of Figure 3 at the bottom of this document.

R2C3: *Figure 4: 1. I recommend restructuring the figure legend so that each panel “(a), (b), etc.” is described separately. For each heatmap, the legend should clearly state the main variable displayed, the numerical range or color scale, and the main findings or notable patterns. Moreover, heatmaps can be improved through the addition of markers (e.g., bold boxes, stars) to identify top contributors or outliers.*

Thank you for your comment on Figure 4. In response to your suggestion, we have now added subplot panels grouping the four sections of the figure by their common colorbars. With this change, we can now more clearly state the variables displayed and the numerical range. For your point on the main findings and notable patterns, we opted to address the main findings and notable patterns of the figure in the main text instead of in the caption, e.g.:

(394-397)

For example, in SSP3-45, the exchange between Europe and Africa is relatively balanced: the prior contributes 48% of the total co-benefits exchanged and the latter, 52% (Figure 5a). Transitioning to more sustainable socioeconomic development leads to Africa – the region with the weaker GDP today – contributing a clear majority of co-benefits to the exchange (72%).

In response to the final suggestion to identify top contributors in the figure we have added white stars to indicate contribution relationships that are within the top 5% highest contributions for

each heatmap. For your reference, we have included the updated version of Figure 4 at the bottom of this document.

R2C4: *Figure 5: 1. The font size for country names and axis labels is too small. Please increase font size and consider bolding labels for clarity; 2. Please add explicit in-panel legends explaining the meaning of black/white dots and arrow directions, rather than relying solely on the main caption.*

Thank you for your comments on Figure 5. In response to these comments, we have increased the font size for the country names and country labels and bolded the text to improve readability. We have also added explicit in-panel legends to explain the scenarios that the black and white dots represent and the regions that the shading indicates. For your reference, we have included the updated version of Figure 5 at the bottom of this document.

2.3 Reviewer #3

R3C0: *I co-reviewed this manuscript with one of the reviewers who provided the listed reports. This is part of the Nature Communications initiative to facilitate training in peer review and to provide appropriate recognition for Early Career Researchers who co-review manuscripts.*

Thank you again for your contributions to this manuscript. We appreciate the time that you have taken to review our manuscript for a second time.

2.4 Reviewer #4

R4C0: *The authors have made thorough and thoughtful revisions in response to the initial round of comments. The manuscript has been significantly improved through the incorporation of additional analyses, clarification of methodologies, and enhanced discussion of results and implications. The introduction of “transboundary fractions” and “exchanges” as well as the inclusion of uncertainty analyses strengthen the paper’s contributions to understanding how climate action may affect international air pollution inequalities. Despite these substantial improvements, several issues require further attention to fully improve the manuscript’s quality.*

We thank you for your positive comments and detailed suggestions on our revised manuscript. We have endeavored to address these concerns below.

R4C1: *The focus on 2040 is well-justified, but the analysis remains a single-time-slice comparison. The supplemental analysis of 2030 is appreciated, but the manuscript should more explicitly discuss the limitations of not capturing temporal evolution, especially since emission pathways and socioeconomic assumptions evolve complexly. A brief discussion on how key inequalities might trend over time (e.g., towards 2050) would strengthen the manuscript.*

Thank you for raising this comment on the temporal evolution of inequalities. We agree that more care is needed when discussing our results given that they are tied to just a single year whereas the inequalities associated with air pollution exchange will naturally evolve over time. To address this comment, we have three responses. First, we have performed an additional analysis for 2050 and now compare the trends in transboundary fractions at the regional level from 2030 to 2050 in a new supplemental figure (Figure S6). For your reference, we have included Figure S6 at the bottom of this document. Second, we have now updated the paragraph after Figure 2 to discuss these new results

[364-374]

The main results presented are for 2040; however, we additionally calculate the differences in transboundary fractions and co-benefits between the base year of study (2040) and 2030 (Figure S4 and S5) across SSP1-26 and SSP3-26. We find that in 2040 transboundary fractions are moderately higher in China, India, the US, and Brazil and lower in parts of Africa compared to SSP1 in 2030; however, these interannual changes are smaller than those attributable to socioeconomic changes. We additionally project changes in transboundary fractions from 2030 to 2050 across the set of scenarios (Figure S6) to identify regional-scale trends. Generally, transboundary fractions remain stable through time; however, in more equal SSPs (i.e., SSP1), transboundary fractions for Africa generally decrease from 2030 to 2050 (e.g., by 15% from 2030 to 2050 in SSP1-45) compared to less equal SSPs (i.e., SSP3) where transboundary fractions increase (e.g., increase by 30% from 2040 to 2050 in SSP3-45).

Lastly, we modified the discussion section of the manuscript to more explicitly call out the limitations of our focus on a single year:

[670-684]

Given that the SSP-RCP emissions are designed to peak at different points across different countries and regions the results presented in this work should be considered only as a snapshot of the inequalities in the year 2040 as the transboundary fractions and exchanges will differ at different points in the future. In this study we do not explore the evolution of inequalities in detail; however, a benefit of this adjoint methodology is its ability to rapidly assess different years – as well as different scenarios – offline without additional simulations. To demonstrate this, we include figures for an additional year (i.e., 2030) in the supplement (Figure S10-S13). Additionally, we compare the transboundary fractions and co-benefits in 2030 to those in 2040 (Figure S4 and S5) at the national-scale and estimate the regional trends in transboundary fractions from 2030 to 2050 (Figure S6); these additional results suggest that less equal SSPs likely will exacerbate transboundary fractions through time than more equal ones. Lastly, although we estimate uncertainty associated with the health calculation, there is additional uncertainty in the projected changes to population, age distribution, and baseline disease rates that are not captured in our analysis along with uncertainty in the air quality modeling that we discuss in detail in the methods.

R4C2: *The revised metrics (transboundary fractions and exchanges) are much clearer now. I suggest the authors present some detailed discussions about their implications for climate justice or policy design. The authors might also more explicitly link metric values to equitable policy outcomes, e.g., whether certain values indicate need for compensation, cooperation, or independent action.*

We are glad to hear that the revised metrics are easier to interpret. We agree that we can further explore the implications of the metrics for policy design. We have taken your suggestion and developed a new figure (Figure S3) to link the transboundary fraction values to specific policy concerns through considering an additional dimension of HDI. This figure has been included at the end of this document. We now discuss this supplemental figure in a new paragraph after Figure 3 and specifically call out values that imply compensation, cooperation, and independent action:

[314-322]

... it is important to consider that transboundary fractions can indicate equitable or inequitable policy outcomes when considered alongside the HDI of a country (Figure S3). Generally, countries with high transboundary fractions require more compensatory actions as they have less control of their air pollution burden, whereas countries with low transboundary fractions can adopt action more independently. However, given that geography and population density influence transboundary fractions, it is crucial to consider a dimension of development alongside this exchange. The greatest targets for compensatory action are those countries with high transboundary fractions and a low level of development. These countries will likely bear heavier health burdens in the future as industrialization and technological progress occur.

We explored this in the earlier manuscript with Figure 5; however, we did not clearly articulate the implications of the results in this figure. We have modified the text to more clearly lay this out:

[415-422]

In our analyses, we consider exchanges in three ways: (1) in their absolute sense an “exchange” represents how emission reduction in one country induces co-benefits in another, (2) as “net exchanges” by differencing the two exchanges in a pair, and (3) the proportion of co-benefits contributed by one member of a pair to the total gross co-benefits exchanged between the pair to determine if a specific country dominates the exchange of air pollution co-benefits. In the latter approach, we note that exchanges in which there is a greater disparity – especially when the low HDI member of the pair is contributing more than the higher one – are key targets for compensatory action.

R4C3: *Line 22. Abstract should distinguish between background and original results. To improve clarity of the abstract, please explicitly introduce key results with phrases such as "our results reveal that..." or "we find that..."*

Thank you for this comment, we have now updated the abstract to more clearly distinguish our results from other studies:

[18-34]

Climate action redresses poor air quality and its health burden by reducing sources that co-emit greenhouse gases and air pollutants such as PM_{2.5}, yet ill-conceived mitigation could induce imbalances in the distribution and exchange of air pollution across international borders. Despite its potential to endanger equality, the future effects of climate action on transboundary PM_{2.5} are largely unstudied. We find that air quality co-benefits are maximized through sustainable development towards RCP-19 (1.32 million deaths avoided) and minimized in fragmentation trends towards RCP-60 (0.32 million). Our results suggest that futures with stronger climate mitigation necessitate more aggressive climate action from developed nations that increases the fraction of co-benefits that are transboundary for Africa, ranging from 8% higher in SSP1 to 53% in SSP3. Additionally, we show that countries with lower human development indices have a greater fraction of their co-benefits originate externally (0.76 in SSP1-26) than more developed countries (0.65) which positions developing nations as more dependent on regional and global climate action than developed countries. We also find that co-benefit exchanges between countries are non-uniform, which suggests country-specific analyses are necessary to develop climate policy that minimize transboundary effects. Our results support the need for nuanced policy design that considers how transboundary co-benefits respond to regional differences, distinct socioeconomic trends, and mitigation strategies.

R4C4: *Line 63-64. The sentence on atmospheric transport and trade is somewhat abrupt. Consider integrating it more smoothly or expanding with a phrase like “In addition to atmospheric transport, trade imbalances also contribute...”*

Thank you for this suggestion, we have now updated this sentence:

Before:

(63-64)

Atmospheric transport drives some of this exchange; however, trade imbalances in consumption and production contribute to air pollution inequalities as well.

After:

[67-69]

In addition to atmospheric transport, trade imbalances in consumption and production also contribute to air pollution inequalities^{19,20}

R4C5: *Line 148-160. Please clarify whether this fraction is calculated based on mass of PM_{2.5} or health impacts (deaths).*

These fractions refer to health impacts (deaths). We have now clarified this in the text:

[168-169]

Transboundary fractions are calculated from the health impacts associated with PM_{2.5}-exposure.

R4C6: *Line 251-261. The correlation between transboundary fraction and HDI is insightful. Have the authors considered also controlling for geographic size or population density? Smaller or coastal countries may have inherently different transboundary fractions regardless of development status.*

You raise a good point as there are a number of variables that drive this correlation (geographic size, population density, prevailing wind direction, how isolated major population centers are from other countries). We explored controlling for population density; however, ultimately settled on performing this analysis using the total health impacts as these are the most policy-relevant number. We note that we discussed this briefly in the discussion:

(476-479)

This proportionally weaker transboundary influence in Asia is a product of population density: major populations in Africa and Europe are in closer proximity to extraregional polluters, whereas in Asia most of the high population is concentrated in the Eastern part of the continent, away from other regions.

However, we agree that it would be useful to comment on some of the overarching trends in more detail. We have added text after Figure 2 to address this:

[275-280]

There are several underlying factors driving the transboundary fractions: e.g., proximity to major polluters, geographic size, prevailing winds, and population density. Island nations, and those isolated by geographic features (e.g., Chile) tend to have more domestic pollution than nations proximate to high polluters (e.g., much of Europe). In the Northern Hemisphere, prevailing westerlies lead to more pollution transport to the East of major polluters in comparison to the West. Thus, it is important to consider that transboundary fractions...

We then tie this into the earlier discussion in response to R4C2.

R4C7: *Line 303-308. The comparison between 2030 and 2040 is useful. Please include in the main text whether the differences are statistically significant or within uncertainty bounds.*

We have now added this to the main text:

[XXX]

We note that our estimates of co-benefits for 2030 lie outside of the lower-bound health estimate for 2040 indicating that the difference in these estimates is not captured by uncertainty in the health calculation alone.

R4C8: *Line 530–553. The uncertainty paragraphs in the Methods and Discussion are somewhat repetitive. Consider rephrasing or cross-referencing to avoid duplication.*

We have now reworded these paragraphs to be more concise and added quantitative examples to the uncertainty analysis in response to R1C1.

R4C9: *Figure 3. The maps are effective, but I suggest consider adding a panel showing the absolute number of transboundary co-benefits (deaths avoided) in addition to the fraction. This would help distinguish between high-fraction but low-impact regions vs. high-impact regions*

Thank you for this suggestion. We agree that a figure presenting the absolute co-benefits would be useful for comparison against the transboundary fractions. We initially experimented with adding this as a new panel to Figure 3 but preferred it as a separate figure, so we have now added this to the supplement (Figure S2) that we include at the bottom of this document. We additionally now point to this figure in the main text:

[271-273]

The total co-benefit numbers are also included (Figure S2) to distinguish areas of high and low impact when interpreting the transboundary fractions.

R4C10: *The authors have done a great job revising the manuscript and addressing most of the initial concerns. The paper presents a valuable and timely analysis of how climate action may affect international air pollution inequalities. With the additional clarifications and discussions suggested above, the manuscript will be suitable for publication in Nature Communications.*

Thank you again for the positive comments and your useful suggestions.

3.0 Tables and Figures

Figure 2. (a) The total deaths avoided for each of the SSP-RCP scenarios relative to SSP3-Baseline in 2040 ordered from the most to least equitable scenarios. Colors indicate the receptor region in which the co-benefits occurred (i.e., where the deaths were avoided). Socioeconomic trends and mitigation strategies are included on the y-axis. Error bars refer to the lower and upper bound uncertainty from the health impact assessment for the total co-benefits. (b) The co-benefits specifically attributable to external action for each of the receptor regions (i.e., the transboundary co-benefits). The fraction of co-benefits that are transboundary broken down by (c) socioeconomic development type and (d) mitigation strategy.

Figure 3. (a) Transboundary fractions for individual countries in a sustainable and strong climate mitigation scenario (SSP1-26), (b) the relative change in transboundary fractions associated with a weaker RCP, i.e., from SSP1-26 to SSP1-45; red values indicate where transboundary fractions have increased whereas blue values indicate decreases, (c) the relative change in transboundary fractions from a less equal SSP, i.e., from SSP1-26 to SSP3-26, (d) the relative change in transboundary fractions from both a weaker RCP and less equal SSP, i.e., from SSP1-26 to SSP3-45.

Figure 4. Exchanges (EXC) of climate co-benefits within and between (a) Europe and Africa, for SSP1-26 in 2040 (left). Contributions are indicated through a logarithmic colormap ranging from 1 to 1000 deaths avoided. Darker colors indicate greater co-benefits in the receptor country attributable to emission reductions in the source country; self-contributions (i.e., the diagonal) are excluded. (b) EXC within Asia and between Asia and Africa; here contributions are indicated in a logarithmic colormap that ranges from 10 to 10000 deaths avoided. (c) Heatmaps of transboundary exchanges of climate action within and between Europe and Africa in 2040 for the scenarios SSP2-26, SSP3-26, SSP4-26, and SSP5-26 relative to SSP1-26. The colormaps are linear and range from -60 to +60 fewer or more deaths avoided compared to SSP1. (d) Heatmaps of transboundary exchanges of climate action within Asia and between Asia and Africa in 2040 for the scenarios SSP2-26, SSP3-26, SSP4-26, and SSP5-26 relative to SSP1-26. The colormaps are linear and range from -120 to +120 fewer or more deaths avoided compared to SSP1. For all subplots, interregional exchanges (i.e., Africa to Europe and Asia to Africa) are provided in an absolute sense – not relative to SSP1. White asterisks are placed to indicate the top 5% highest (absolute) values for each heatmap.

Figure 5. The change in the percentage of co-benefits exchanged ($TEC \times 100\%$) between (a) regional and (b) country pairs that is contributed by the higher GDP (first name) from SSP3-45 to SSP1-45 in 2040. For the regional exchanges, the arrow points from the less equal scenario (SSP3-45) to the more equal scenario (SSP1-45) and the black dot represents SSP3-45 and the white dot represents SSP1-45. For the country exchanges, the arrow and dots are the same but the background shading indicates in which region the exchange occurs as labelled in the legend below the subplots; grey indicates exchange between different regions. Panels (c) and (d) are the same as (a) and (b), respectively, but for middle-of-the-road development (SSP2 vs SSP1) instead of fragmentation development.

Figure S2. The country-level total co-benefits in SSP1-26 (a) and the relatively fewer co-benefits in a weaker RCP (SSP1-45) (b), a less equal SSP (SSP3-26) (c), and both a weaker RCP and less equal SSP (SSP3-45) (d). The co-benefits in (a) are presented on a log-scale colormap; all other panels are presented on a linear colormap.

Figure S3. Policy scorecard for determining whether a specific scenario had minimal, medium or major equity concerns based on the HDI value and transboundary fraction of the receptor country. Example values for SSP1-19 (Sustainability) and SSP3-60 (Fragmentation) are included for nine countries to indicate different values and how changing scenarios affect equity.

Figure S6. Regional estimates of transboundary fractions for the six major regions for all scenarios considered in this study identified by rows (for SSPs) and columns (for RCPs) across 2030, 2040, and 2050. For SSP3-34, SSP3-45, and SSP3-60, 2030 data are excluded as these scenarios were nearly identical to the baseline scenario.